# THE HIDDEN LABEL-MARGINAL BIASES OF SEGMENTATION LOSSES

## ABSTRACT

Most segmentation losses are arguably variants of the Cross-Entropy (CE) or Dice losses. In the abundant segmentation literature, there is no clear consensus as to which of these losses is a better choice, with varying performances for each across different benchmarks and applications. In this work, we develop a theoretical analysis that links these two types of losses, exposing their advantages and weaknesses. First, we provide a constrained-optimization perspective showing that CE and Dice share a much deeper connection than previously thought: They both decompose into label-marginal penalties and closely related ground-truth matching penalties. Then, we provide bound relationships and an information-theoretic analysis, which uncover hidden label-marginal biases: Dice has an intrinsic bias towards specific extremely imbalanced solutions, whereas CE implicitly encourages the ground-truth region proportions. Our theoretical results explain the wide experimental evidence in the medical-imaging literature, whereby Dice losses bring improvements for imbalanced segmentation. It also explains why CE dominates natural-image problems with diverse class proportions, in which case Dice might have difficulty adapting to different label-marginal distributions. Based on our theoretical analysis, we propose a principled and simple solution, which enables to control explicitly the label-marginal bias. Our loss integrates CE with explicit $\mathcal{L}_1$ regularization, which encourages label marginals to match target class proportions, thereby mitigating class imbalance but without losing generality. Comprehensive experiments and ablation studies over different losses and applications validate our theoretical analysis, as well as the effectiveness of our explicit label-marginal regularizers.

## 1 INTRODUCTION

Semantic segmentation is one of the most investigated problems in computer vision, and has been impacting a breadth of applications, from natural-scene understanding (Cordts et al., 2016; Kirillov et al., 2019) to medical image analysis (Litjens et al., 2017; Dolz et al., 2018). In the recent years, deep learning methods have dominated the field, as a result of the great capacity of Convolutional Neural Networks (CNN) (He et al., 2016) to automatically learn representations from large-scale data sets (Long et al., 2015; Ronneberger et al., 2015; Zhao et al., 2017; Chen et al., 2018; Yuan et al., 2020). Semantic segmentation is often stated as a pixel-wise classification task, following the optimization of a loss function expressed with summations over the ground-truth regions, as in the standard Cross-Entropy (CE) loss. A relevant aspect of segmentation problems is class imbalance, *i.e.*, unequal proportions of the segmentation regions, which may cause large-region terms in the objective to completely dominate small-region ones. A representative example is the popular Cityscapes dataset (Cordts et al., 2016), where the average proportions of some classes, such as *motorcycle* or *bicycle*, are below $1\%$. In these scenarios, besides specifically designed CNN architectures or training schemes (Tao et al., 2020; Bao et al., 2021), the loss function to be minimized during learning plays a critical role, and has triggered a large body of research works in the last years (Lin et al., 2017; Wong et al., 2018; Kervadec et al., 2021b; Milletari et al., 2016; Sudre et al., 2017; Kervadec et al., 2021a).

There exists a great diversity of loss functions for image segmentation, which can be categorized into two main families, and are arguably variants of CE, the Dice loss (Ma et al., 2021; Yeung et al., 2021), or combinations of both (Wong et al., 2018; Taghanaki et al., 2019). The first family is mo-

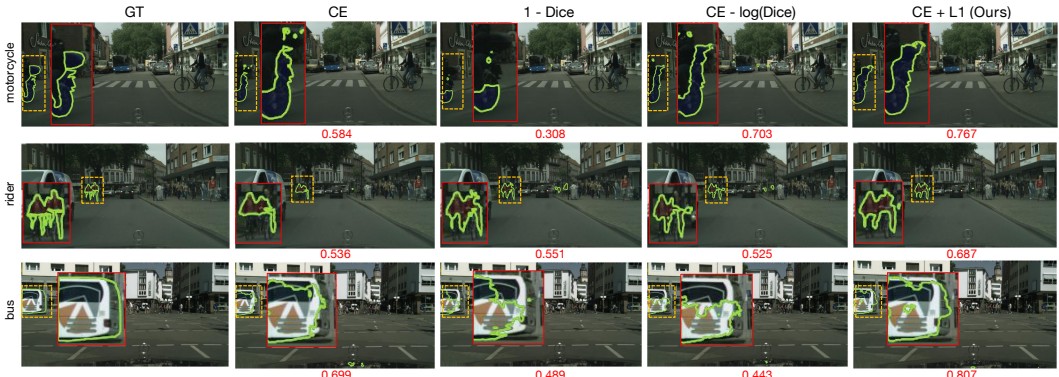

Figure 1: **The visual improvement of our solution compared to various losses under different classes in Cityscapes.** The ground-truth is given in the left-most column. A magnified contour of the segmentation is also provided alongside the original target within each image. The numeric below each image indicates the IoU score for the corresponding prediction result. The examples show how Dice has a bias towards small regions, while CE is more stable for different scenarios. Our solution is capable of leveraging the best aspects of both. See Sec. 2 for the explicit theoretical analysis and Sec. 3 for the detailed experimental results. Best seen in color.

tivated by distribution measures, *i.e.*, CE and its variants, and is directly adapted from classification tasks. To deal with class imbalance, various extensions of CE have been investigated, such as increasing the relative weights for minority classes (Ronneberger et al., 2015), or modifying the loss so as to account for performance indicators during training, as in the popular Focal loss (Lin et al., 2017) or TopK loss (Wu et al., 2016). The second main family of losses is inspired by geometrical metrics. In this category, the most popular losses are linear Dice (Milletari et al., 2016) and its extensions, such as the logarithmic (Wong et al., 2018) or generalized (Sudre et al., 2017) Dice loss. Borrowing the idea of the weighted CE, the latter introduces class weights to increase the contributions of the minority classes. These loss functions are motivated by the geometric Dice coefficient, which measures the overlap between the ground-truth and predicted segmentation regions.

In the literature, to our knowledge, there is no clear consensus as to which category of losses is better, with the performances of each varying across data sets and applications. It has been empirically argued that the Dice loss and its variants are more appropriate for extreme class imbalance, and such empirical observations are the main motivation behind the wide use and popularity of Dice in medical-imaging applications (Milletari et al., 2016; Jha et al., 2020). CE, however, dominates most recent models in the context of natural images (Chen et al., 2018; Zhao et al., 2017; 2018; Yuan et al., 2020). Therefore, beyond experimental evidence, there is a need for a theoretical analysis that clarifies which segmentation loss to adopt for a given task, a decision that affects performance significantly.

On the surface, these two categories of losses (*i.e.*, geometry-based vs. distribution-based) seem unrelated. Furthermore, an important body of work in the literature suggests that CE and Dice are complementary, which has motivated composite losses integrating both, *e.g.*, (Wong et al., 2018; Taghanaki et al., 2019). Such composite losses perform very competitively in extremely imbalanced segmentation, as shown by (Wong et al., 2018; Taghanaki et al., 2019), among several other recent works (Isensee et al., 2018; Ma et al., 2021).

In this paper, we provide a constrained-optimization perspective showing that, in fact, CE and Dice share a much deeper connection than previously thought: They both decompose into label-marginal penalties and closely related ground-truth matching penalties. Our theoretical analysis highlights encoded hidden label-marginal biases in Dice and CE, and shows that the main difference between the two types of losses lies essentially in those label-marginal biases: Dice has an intrinsic bias preferring very small regions, while CE implicitly encourages the right (ground-truth) region proportions. Our results explain the wide experimental evidence in the medical-imaging literature, whereby using or adding Dice losses brings improvements for imbalanced segmentation with extremely small regions. It also explains why CE dominates natural-image problems with diverse class proportions, in which case Dice might have difficulty adapting to different label-marginal distributions (see Table

3 and the examples depicted in Fig. 1). Based on our theoretical analysis, we propose principled and simple loss functions, which enable to control explicitly the label-marginal bias. Our solution integrates the benefits of both categories of losses, mitigating class imbalance but without losing generality, as shown in the examples in Fig. 1.

Our contributions are summarized as follows:

- Showing through an explicit bound relationship (Proposition 1) that the Dice loss has a hidden label-marginal bias towards specific extremely imbalanced solutions, preferring small structures, while losing the flexibility to deal effectively with arbitrary class proportions.
- Providing an information-theoretic perspective of CE, via Monte-Carlo approximation of the entropy of the learned features (Proposition 2). This highlights a hidden label-marginal bias of CE, which encourages the proportions of the predicted segmentation regions to match the ground-truth proportions.
- Introducing new loss functions to control label-marginal biases: Our losses integrate CE with explicit regularization terms based on $\mathcal{L}_1$ or the KL divergence, which encourage label marginals to match target class proportions.
- Comprehensive experiments and ablation studies over different losses and applications, including natural and medical-imaging data, validate our theoretical analysis, as well as the effectiveness of our explicit label-marginal regularizers.

## 2 FORMULATION

Table 1: **Notations, formulations and approximations used in this paper.** $\mathcal{F}$ and $\mathcal{K}$ denotes the random variables associated with the learned features and the labels, respectively. $\mathbb{P}$ denotes probability. $|.|$ denotes cardinality when the input is a set and the standard absolute value when the input is a scalar. Note that network parameters $\theta$ are omitted in the prediction quantities, so as to simplify notations, as this does not lead to ambiguity.

| Dataset | |
|---|---|
| Concept | Formula |
| Indices/number of classes | $1 \leq k \leq K$ |
| Spatial image domain | $\Omega \subset \mathbb{R}^2$ |
| Labels of pixel $i \in \Omega$ | $y_{ik} \in \{0, 1\}$ |
| GT region $k$ | $\Omega_k = \{i \in \Omega \mid y_{ik} = 1\}$ |
| GT proportion of region $k$ | $\hat{y}_k = \frac{|\Omega_k|}{|\Omega|}$ |
| GT label-marginal prob. | $\mathbf{y} = (\hat{y}_k)_{1 \leq k \leq K}$ |

| Modeling | |
|---|---|
| Concept | Formula |
| Model parameters | $\theta$ |
| Feature embedding at pixel $i \in \Omega$ | $\mathbf{f}_i^\theta$ |
| Softmax predictions at pixel $i \in \Omega$ | $p_{ik} = \mathbb{P}(k \mid \mathbf{f}_i^\theta)$ |
| Predicted proportion of class $k$ | $\hat{p}_k = \frac{1}{|\Omega|} \sum_{i \in \Omega} p_{ik}$ |
| Predicted label-marginal prob. | $\mathbf{p} = (\hat{p}_k)_{1 \leq k \leq K}$ |
| $(K-1)$-simplex | $\Delta_K = \{\mathbf{p} \in [0,1]^K \ / \ \sum_k \hat{p}_k = 1\}$ |

| Losses, label-marginal regularizers and information-theoretic quantities | |
|---|---|
| Concept | Formula |
| Weighted cross-entropy | $\text{CE} = -\sum_{k=1}^K \frac{1}{|\Omega_k|} \sum_{i \in \Omega_k} \log(p_{ik})$ |
| Dice coefficient for region $k$ | $\text{Dice}_k = \frac{2 \sum_{i \in \Omega_k} p_{ik}}{\sum_{i \in \Omega} p_{ik} + |\Omega_k|}$ |
| Label-marginal KL divergence | $\mathcal{D}_{\text{KL}}(\mathbf{y} \| \mathbf{p}) = \sum_{k=1}^K \hat{y}_k \log(\frac{\hat{y}_k}{\hat{p}_k})$ |
| Label marginal $\mathcal{L}_1$ distance | $\mathcal{L}_1(\mathbf{y}, \mathbf{p}) = \sum_{k=1}^K |\hat{y}_k - \hat{p}_k|$ |
| *Monte-Carlo* estimate of the entropy of features given region $k$ | $\mathcal{H}(\mathcal{F} \mid \mathcal{K} = k) \approx -\frac{1}{|\Omega_k|} \sum_{i \in \Omega_k} \log(\mathbb{P}(\mathbf{f}_i^\theta \mid k))$ |

Semantic segmentation is often stated as a pixel-wise classification task, following the optimization of a loss function for training a deep network. In Table 1, we present the notations, formulations and approximations used in our subsequent discussions. Besides the basic notations of the task (such as networks predictions), we explicitly include the loss functions, label-marginal regularizers and information-theoretic quantities that will be discussed in the following sections. We note that, to facilitate the reading of our analysis, we write the CE and Dice losses in a non-standard way using summations over the ground-truth segmentation regions, rather than as functions of the labels. Also, while we provide the CE loss for all segmentation regions, we give Dice for a single region. This is to accommodate two variants of the Dice loss in the literature: in the binary case, Dice is typically

used for the foreground region only (Milletari et al., 2016); in the multi-region case, it is commonly used over all the regions (Wong et al., 2018). Finally, to simplify notation, we give all the loss functions for a single training image, without summations over all training samples (as this does not lead to any ambiguity, neither does it alter the analysis hereafter). In the training iterations, we use the mean values across all the training samples via standard mini-batch optimization.

## 2.1 DEFINITION OF LABEL-MARGINAL BIASES AND PENALTY FUNCTIONS

In the following, we analyse the label-marginal biases inherent to CE and Dice losses, and show that the main difference between the two types of losses lies essentially in those label-marginal biases. To do so, we provide a constrained-optimization perspective of the losses: We define a label-marginal bias as a soft *penalty* function for the hard equality constraint $\mathbf{p} = \mathbf{t}$, where $\mathbf{t}$ is a given (fixed) target distribution. Such a penalty encourages the predicted label-marginal $\mathbf{p}$ to match a given target distribution $\mathbf{t}$. In the general context of constrained optimization, penalty functions are widely used (Bertsekas, 1995). In general, penalty methods replace equality constraints of the form $\mathbf{p} = \mathbf{t}$ by adding a term $g(\mathbf{p})$ to the main objective being minimized. Such a penalty function $g$ increases when $\mathbf{p}$ deviates from target $\mathbf{t}$. By definition, for the constraint $\mathbf{p} = \mathbf{t}$, with the domain of $\mathbf{p}$ being probability simplex $\Delta_K$, a penalty $g(\mathbf{p})$ is a continuous and differentiable function, which reaches its global minimum when the constraint is satisfied, *i.e.*, it verifies: $g(\mathbf{t}) \leq g(\mathbf{p}) \, \forall \mathbf{p} \in \Delta_K$.

## 2.2 THE LINK BETWEEN CROSS ENTROPY AND DICE

To ease the discussion in what follows, we will start by analyzing the link between CE and the logarithmic Dice, along with the label-marginal bias of the latter (Proposition 1). Then, we discuss a bounding relationship between the different Dice variants. Finally, we will provide an information-theoretic analysis, which highlights the hidden label-marginal bias of CE (Proposition 2).

Let us consider the logarithmic Dice loss in the multi-class case. This loss decomposes (up to a constant) into two terms, a ground-truth matching term and a label-marginal bias:

$$-\sum_{k=1}^{K} \log(\text{Dice}_k) \overset{\text{c}}{=} \underbrace{-\sum_{k=1}^{K} \log\left(\frac{1}{|\mathbf{\Omega}_k|} \sum_{i \in \mathbf{\Omega}_k} p_{ik}\right)}_{\text{Ground-truth matching: DF}} + \underbrace{\sum_{k=1}^{K} \log\left(\hat{p}_k + \hat{y}_k\right)}_{\text{Label-marginal bias: DB}} \tag{1}$$

where $\overset{\text{c}}{=}$ stands for equality up to an additive and/or non-negative multiplicative constant. The ground-truth matching term in Eq. (1) is a lower bound on the cross-entropy loss (CE) due to Jensen's inequality and the convexity of function $-\log(x)$: DF $\leq$ CE. Therefore, minimizing CE could be viewed as a proxy for minimizing term DF that appears in the logarithmic Dice. In fact, from a constrained-optimization perspective, DF and CE are very closely related and could be viewed as two different penalty functions enforcing the same equality constraints: $p_{ik} = 1, \forall i \in \mathbf{\Omega}_k, \forall k$. Both DF and CE are monotonically decreasing functions of each softmax and reach their global minimum when these equality constraints are satisfied. Therefore, they encourage softmax predictions $p_{ik}$ for each region $\mathbf{\Omega}_k$ to reach their target ground-truth values of 1. Of course, this does not mean that penalties CE and DF yield exactly the same results. The difference in the results that they may yield is due to the optimization technique (*e.g.*, different gradient dynamics in the standard training of deep networks as the penalty functions have different forms).

## 2.3 THE HIDDEN LABEL-MARGINAL BIAS OF DICE

The following proposition highlights how the label-marginal term DB in Eq. (1) encourages specific extremely imbalanced solutions.

**Proposition 1.** *Let* $\mathbf{t} = \left(\hat{t}_j\right)_{1 \leq j \leq K} \in \{0, 1\}^K$ *denote the simplex vertex verifying:* $\hat{t}_j = 1$ *when* $\hat{y}_j = \max_{1 \leq k \leq K} \hat{y}_k$ *and* $\hat{t}_j = 0$ *otherwise. For variables* $\mathbf{p} = (\hat{p}_k)_{1 \leq k \leq K}$ *and fixed distribution* $\mathbf{y} = (\hat{y}_k)_{1 \leq k \leq K}$, *the label-marginal term in Eq. (1) reaches its minimum over the simplex at* $\mathbf{t}$:

$$\sum_{k=1}^{K} \log\left(\hat{t}_k + \hat{y}_k\right) \leq \sum_{k=1}^{K} \log\left(\hat{p}_k + \hat{y}_k\right) \quad \forall \mathbf{p} \in \Delta_K \tag{2}$$

*Proof.* The details of the proof are deferred to Appendix A.1. The main technical ingredient is based on Jensen's inequality and the concavity of penalty DB with respect to simplex variables $\mathbf{p}$. □

Inequality (2) means that the label-marginal term in Dice in Eq. (1) is a penalty function for constraint $\mathbf{p} = \mathbf{t}$, where $\mathbf{t}$ is the simplex vertex given in Proposition 1. Therefore, it encourages extremely imbalanced segmentations, where a specific region includes all the pixels and the remaining regions are empty. All in all, the logarithmic Dice loss integrates a hidden label-marginal prior preferring extremely imbalanced segmentations, which is optimized jointly with a ground-truth matching term similar to CE. It is worth noting that, in the two-class (binary) segmentation case, Dice might be used for the foreground region only, as in the popular work in (Milletari et al., 2016), for instance. Similarly to the multi-class case discussed above, a single Dice term also decomposes into a ground-truth matching term and label-marginal penalty, with the latter encouraging extremely imbalanced binary segmentations. We provide more details for this case in Appendix B.

## 2.4 ON THE LINK BETWEEN THE DIFFERENT VARIANTS OF DICE

The label-marginal analysis we discussed above is based on the standard logarithmic Dice loss. Here, we argue that both logarithmic and linear Dice are very closely related and, hence, the linear Dice also hides a class-imbalance bias. In fact, from a constrained-optimization perspective, the two losses could be viewed as different penalty functions for imposing constraints: $\text{Dice}_k = 1 \, \forall k$. Both functions $-\log(x)$ and $(1-x)$ are monotonically decreasing in $[0, 1]$ and achieve their minimum in $[0, 1]$ at $x = 1$. Furthermore, the logarithmic Dice is an upper bound on the linear one. This follows directly from: $-\log(t) \geq 1 - t \quad \forall t > 0$. Of course, this does not mean that optimizing these two variants leads to exactly the same results. The differences in their results might be due to optimization (*i.e.*, different gradient dynamics stemming from logarithmic and linear penalties).

## 2.5 THE HIDDEN LABEL-MARGINAL BIAS OF CE

In the following, we give an information-theoretic perspective of CE, via a generative view of network predictions and a Monte-Carlo approximation of the entropy of the learned features given the labels. This highlights a hidden label-marginal bias of CE, which encourages the proportions of the predicted segmentation regions to match the ground-truth proportions.

**Proposition 2.** *Let $\mathcal{F}$ and $\mathcal{K}$ denote the random variables associated with the learned features and the labels, respectively, and $\mathcal{H}(\mathcal{F}|\mathcal{K})$ the conditional entropy of learned features given the labels, estimated via Monte-Carlo :*

$$\mathcal{H}(\mathcal{F}|\mathcal{K}) \approx \sum_{k}^{K} y_k \mathcal{H}(\mathcal{F}|\mathcal{K} = k) \approx -\frac{1}{|\mathbf{\Omega}|} \sum_{k}^{K} \sum_{i \in \mathbf{\Omega}_k} \log(\mathbb{P}(\mathbf{f}_i^\theta | k)) \tag{3}$$

*where $\mathcal{H}(\mathcal{F}|\mathcal{K} = k)$ is the empirical estimate of the conditional entropy of features given a specific class $k$ (expression in Table 1) and $\mathbb{P}(\mathbf{f}_i^\theta | k)$ denotes the probability of the learned features given class $k$. We have the following generative view of CE:*

$$\text{CE} \overset{\text{c}}{=} \underbrace{\mathcal{H}(\mathcal{F}|\mathcal{K})}_{\textit{Ground-truth matching}} + \underbrace{\mathcal{D}_{\text{KL}}(\mathbf{y}||\mathbf{p})}_{\textit{Label-marginal bias}} \tag{4}$$

The detailed proof is deferred to Appendix A.2. The approximation of $\mathcal{H}(\mathcal{F}|\mathcal{K} = k)$ in the second line of Eq. (3) is based on the well-known Monte-Carlo estimation (Kearns et al., 1997; Tang et al., 2019). Then the relationship in Eq. (4) follows from Eq. (3), after some manipulations, using Bayes rule $\mathbb{P}(\mathbf{f}_i^\theta | k) \propto \frac{p_{ik}}{\hat{p}_k}$ and $\sum_{i \in \mathbf{\Omega}_k} \log(\hat{p}_k) = |\mathbf{\Omega}_k| \log(\hat{p}_k)$.

This information-theoretic view of CE shows that the latter has an implicit (hidden) label-marginal bias towards the ground-truth region proportions (the KL term). This bias competes with the entropy term, which encourages low uncertainty (variations) within each ground-truth segmentation region $\mathbf{\Omega}_k$. The entropy term could be viewed as a ground-truth matching term: it reaches its global minima when the feature embedding is constant within each region. If used alone, the entropy term may lead to trivial imbalanced solutions. The label-marginal KL term avoids such trivial solutions by matching the ground-truth class proportions. Note that there is no mechanism in CE to control the relative contributions of those two competing terms as they are implicit in CE.

## 2.6 OUR SOLUTION

Our analysis shows that Dice, CE and their combinations, *e.g.*, CE − log(Dice), are closely related and enforce two types of competing constraints : ground-truth matching and label-marginal constraints. However, there is no clear consensus in the literature as to which loss is better, with the performances of each varying across data sets and applications. This variability in performances could be explained by two fundamental factors:

• **The difference in the label-marginal prior**. The label-marginal priors are different as Dice has an intrinsic bias preferring very small regions, while CE encourages the right (ground-truth) region proportions. This might explain the wide experimental evidence in the medical imaging literature, where using or adding Dice losses brings improvements for imbalanced segmentation with extremely small regions.

• **Weighting the contribution of the bias term**. Our analysis suggests that CE should be preferred over Dice in all cases and applications (both balanced/imbalanced segmentation, or segmentation problems with high variability in region proportions) as it promotes the right label-marginal distribution. While this seems to be widely the case in natural image segmentation, where Dice is uncommon, the extensive experimental evidence in the medical-image segmentation literature suggests otherwise, especially in extremely imbalanced problems. We argue that this is due to the relative contribution of the label-marginal term in the overall objective. Controlling such label-marginal contribution is very important in imbalanced problems. In particular, it mitigates the difficulty that the ground-truth matching terms differ by several orders of magnitude across regions, as in CE, which causes large-region terms to completely dominate small-region ones. This analysis also resonates with the fact that combo losses such as CE - $\lambda \log$ (Dice) perform very competitively in imbalanced segmentation, as shown by (Wong et al., 2018; Taghanaki et al., 2019), among several other recent works. In this case, controlling the relative contribution of each of these terms indirectly controls the weight of the label-marginal bias. Note that such control is not possible when using CE alone or Dice alone, as the label-marginal biases in these losses are hidden (implicit).

We propose a principled and simple solution, which enables to control explicitly the label-marginal bias, via regularization losses that encourage the correct class proportions and are used in conjunction with CE :

$$CE + \lambda \mathcal{R}(\mathbf{y}; \mathbf{p}) \qquad (5)$$

Our label-marginal regularizers increase the contribution of the minority classes in imbalanced problems, but, unlike Dice, do not lose adaptability to problems with various class proportions. Our extensive experiments and ablation studies over different losses and applications demonstrate the effectiveness of our explicit label-marginal regularizers. We investigate different forms of regularization, including the $\mathcal{L}_1$ norm, *i.e.*, $\mathcal{R}(\mathbf{y}; \mathbf{p}) = \mathcal{L}_1(\mathbf{y}, \mathbf{p})$, and the KL divergence, *i.e.*, $\mathcal{R}(\mathbf{y}; \mathbf{p}) = \mathcal{D}_{KL}(\mathbf{y}||\mathbf{p})$; see Table 1 for the expressions of $\mathcal{D}_{KL}$ and $\mathcal{L}_1$. In Fig. 2, we depict our different regularizers as functions of the label-marginal distribution for a binary-segmentation case, with the

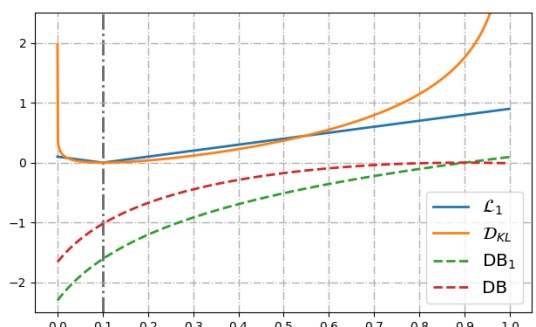

Figure 2: **Different label-marginal biases.** The ground-truth foreground region proportion is set to 0.1. The expression of penalty $DB_1$ is provided in Appendix B, and corresponds to the two-class (binary) variant of Dice, where the loss is used over the foreground region only. Penalty $\mathcal{L}_1$ presents better gradient dynamics at the vicinity of label marginal $\hat{p}_1 = 0$. Best seen in color.

foreground-region proportion set to 0.1, along with the bias terms in Dice. While our $\mathcal{D}_{KL}$ and $\mathcal{L}_1$ regularizers may deliver comparable performances (see the experimental section), $\mathcal{L}_1$ might be a better option for extremely imbalanced segmentations, due to its gradient properties and stability at the vicinity of 0, *i.e.*, when the label-marginal probability $\hat{p}_1$ is close to 0. Notice that, at the vicinity of zero, both first and second derivatives of the regularizer are *unbounded* for $\mathcal{D}_{KL}$, but *bounded* and *constant* for $\mathcal{L}_1$. Our experiments on imbalanced medical image segmentation confirm the effectiveness of the $\mathcal{L}_1$ regularizer.

## 3 EXPERIMENTS

**Datasets.** We report results on both medical and natural image segmentation benchmarks, *i.e.*, *Retinal Lesions* (Wei et al., 2020) and *Cityscapes* (Cordts et al., 2016). Retinal Lesions is a large collection of color fundus images, as shown in Fig. 3 and Fig. G.1. It is noted that a panel of 45 experienced ophthalmologist was formed to label this dataset and each image was assigned to at least three annotators to get trustworthy pixel-level annotations(Wei et al., 2020). In our experiments, we employ its public version[1], containing $1,593$ samples, and conduct our experiments in the binary setting to segment the lesion region versus background. The dataset is randomly divided into training $(70\%)$, validation $(10\%)$ and test $(20\%)$ set, whose images are resized to $512 \times 512$. We tune the hyper-parameters of each model on the validation set, and report the results on the test set. The natural scene dataset, *Cityscapes*, is a large-scale dataset with high quality pixel-level annotations of $5,000$ images across 19 categories, containing both stuff and objects with high variation in the class proportions distribution. We use the official data split, which contains $2,975$ samples for training, $500$ samples for validation and $1,512$ samples for testing. All the input images are resized to $512 \times 1024$ for training, and $1024 \times 2048$ for testing. Fig. C.1 in Appendix C shows the distribution of region proportions for Retinal Lesions and the 19 classes in Cityscapes, highlighting the challenging class imbalance and high diversity on target sizes.

**Backbones.** It is worth stressing that the theoretical discussion in this paper is model-agnostic. Thus, in our experiments we employ Res34-FPN, Res50-FPN (Lin et al., 2017) and Res34-Unet, Res50-Unet (Ronneberger et al., 2015) on Retinal Lesions, while we show results with Res50-FPN and Res101-FPN on Cityscapes, whose implementations are publicly available[2].

**Training.** The parameters of the encoder are initialized with the pre-trained weights on Imagenet, while those in the decoder head are randomly initialized. On Retinal Lesions, we train the model during 60 epochs, with a batch size of 8, via the Adam optimizer. The initial learning rate is set to 1e-4, and halved if the loss on the validation set does not decrease within 5 epochs. On Citycapes, the main setting is adopted from the state-of-the-art library[3]. Specifically, the SGD optimizer is used for training, with an initial learning rate set to $0.01$ and a batch size of 8. During the 100 training epochs, we use an iteration-wise polynomial strategy to linearly scale the learning rate down to a minimum of 1e-4. For all the models, we tune the hyper-parameters on the validation and report the best results on the corresponding test set. Conventional data augmentations, such as random cropping, mirror flipping and changes in brightness, are employed on both datasets.

**Losses.** We evaluate the proposed loss function in Eq. (5) with two regularizing terms, *i.e.*, $\mathcal{D}_{\mathrm{KL}}$ and $\mathcal{L}_1$. In our implementation, we use a modified soft-max function with a temperature parameter when computing the predicted region proportion, as this enables a better estimate of the actual region proportion (refer to Appendix D for details). We compare our results with the following baselines: CE, Focal loss (Lin et al., 2017), Dice related losses, *i.e.*, linear/logarithmic/generalized Dice, and the composite loss combining CE and logarithmic Dice.

**Evaluation metrics.** On Retinal Lesions, we use two standard metrics in medical image segmentation, *i.e.*, Dice Similarity Coefficient (DSC) and modified Hausdorff Distance (HD-95). Particularly, HD-95 represents the 95th percentile of the symmetric Hausdorff Distance (HD) between the binary objects in two images. On Cityscapes, we resort to the standard Intersection-Over-Union (IoU) score, which is widely employed in natural image segmentation.

**Results on Retinal Lesions.** Table 2 reports the quantitative comparison between the proposed loss, with the two different regularizers, and baselines on the Retinal Lesions dataset. Regarless of the backbone network, the proposed $CE+\mathcal{L}_1$ loss consistently outperforms all the others with a DSC score of $54.5\%$ for Res50-FPN and $54.3\%$ for Res50-Unet. Compared to CE, our best model, *i.e.*, $\mathcal{L}_1$ as the label-marginal regularizer, brings nearly $3.0\%$ improvement in terms of DSC and between $5.6\%$ and $8.6\%$ in terms of HD-95, depending on the backbone. Even though these differences are reduced with respect to the best baseline, this gap is still significant, particularly in terms of HD-95. It is noteworthy to mention that while DSC is more sensitive to the internal filling of the target region, the HD-95 is more sensitive to the segmentation boundary. Thus, the proposed loss is more effective

---

[1] `https://github.com/WeiQijie/retinal-lesions`
[2] `https://github.com/qubvel/segmentation_models.pytorch`
[3] `https://github.com/open-mmlab/mmsegmentation`

Table 2: **Quantitative evaluations of different losses on Retinal Lesions.** Average DSC and HD95 values (and standard deviation over three independent runs) achieved on the test set are reported. Note that Dice$_1$ is implemented for all the Dice related losses for the binary setting on this dataset, and DB$_1$ refers to label-marginal bias for the binary Dice (details can be found in Appendix B).

| | Res34-FPN | | Res34-Unet | | Res50-FPN | | Res50-Unet | |
|---|---|---|---|---|---|---|---|---|
| Loss | DSC (%) | HD-95 (mm) | DSC(%) | HD-95 (mm) | DSC(%) | HD-95 (mm) | DSC(%) | HD-95 (mm) |
| CE | 51.4 (0.1) | 85.50 (2.04) | 52.4 (0.5) | 85.46 (3.71) | 52.7 (0.1) | 80.58 (2.63) | 52.7 (0.3) | 84.27 (4.12) |
| Focal loss (Lin et al., 2017) | 51.2 (0.2) | 84.38 (4.86) | 51.2 (0.8) | 88.56 (4.90) | 52.9 (0.4) | 80.87 (2.75) | 51.8 (0.6) | 84.41 (1.93) |
| $1 - \text{Dice}_1$ (Milletari et al., 2016) | 52.0 (0.7) | 84.80 (3.33) | 52.3 (0.7) | 89.19 (2.44) | 52.0 (0.7) | 82.20 (4.01) | 53.2 (0.1) | 81.93 (4.00) |
| $-\log(\text{Dice}_1)$ (Wong et al., 2018) | 51.7 (0.9) | 86.02 (4.44) | 52.7 (0.5) | 85.84 (0.18) | 52.0 (1.0) | 81.69 (3.57) | 53.5 (0.5) | 82.15 (2.76) |
| GDice (Sudre et al., 2017) | 51.9 (0.7) | 86.70 (2.72) | 52.4 (0.5) | 89.99 (1.28) | 53.4 (0.3) | 80.79 (2.36) | 53.7 (0.4) | 86.84 (3.20) |
| $\text{CE} - \log(\text{Dice}_1)$ (Wong et al., 2018) | 52.2 (0.5) | 81.60 (6.70) | 52.6 (0.4) | 84.13 (2.40) | 53.2 (0.5) | 78.38 (2.06) | 53.4 (0.3) | 82.55 (2.11) |
| $\text{CE} + \text{DB}_1$ | 51.7 (0.9) | 82.72 (1.66) | 51.8 (0.2) | 86.07 (1.83) | 53.4 (0.4) | 78.67 (1.18) | 53.2 (0.1) | 83.04 (0.46) |
| $\text{CE} + \mathcal{D}_{\text{KL}}$ (Ours) | 52.7 (0.3) | 83.31 (2.40) | 52.8 (0.3) | 82.66 (2.40) | 53.8 (0.1) | 77.12 (1.82) | 53.8 (0.4) | 80.17 (1.28) |
| **$\text{CE} + \mathcal{L}_1$ (Ours)** | **52.8 (0.3)** | **80.68 (2.49)** | **53.1 (0.2)** | **81.15 (2.75)** | **54.5 (0.2)** | **74.99 (1.96)** | **54.3 (0.2)** | **77.03 (1.51)** |

to predict better boundaries than existing losses. Furthermore, as shown in Fig. C.1 in Appendix C, the average region proportion is considerably low, *i.e.*, $3.8\%$, and varies significantly (the standard deviation is $6.6\%$ with a maximum of $57.8\%$). Under this highly heterogeneous scenario in terms of class region proportions, the proposed losses present a more stable performance across different runs and backbones, which is reflected in their lower variances. This can be explained by their better adaptability to different target sizes and better gradient dynamics at the vicinity of label marginal $\hat{p}_k = 0$ (as shown in Fig. 2). Another interesting finding is that, by combining CE with the bias term of Dice ($\text{CE} + \text{DB}_1$), we obtain results close to combo loss $\text{CE} - \log(\text{Dice}_1)$(Wong et al., 2018). This validates our theoretical insight that the fundamental difference between CE and Dice lies in their distinct hidden biases.

**On the balancing weight in the composite losses.** In this section we evaluate the impact of the balancing weight $\lambda$ of the proposed loss in Eq. (5), which balances the effects of the regularization term, as well as the balancing weight in $\text{CE} - \log(\text{Dice}_1)$. In our experiments, we empirically found that the best $\lambda$ values $for different penalty terms$ are : 1.0 for $\text{CE} + \mathcal{L}_1$, 0.1 for $\text{CE} + \mathcal{D}_{\text{KL}}$, 0.01 for $\text{CE} + \text{DB}_1$. These values are consistent with the given arguments in Sec. 2.6, which emphasize that $\mathcal{L}_1$ has a better gradient evolution for large learning rates and relatively high weighting, especially at the beginning of training. Regarding the widely used combo loss of $\text{CE} - \log(\text{Dice}_1)$, setting its balancing weight to 1.0 yielded consistent performance across the datasets and networks we used. Details of this experimental study can be found in Appendix E. Therefore, with the same hyper-parameter budget, $\text{CE} + \mathcal{L}_1$ performed better than all the other related composite loss functions. Note that we use the best empirical values of $\lambda$ found on Retinal Lesions for the experiments on Cityscapes.

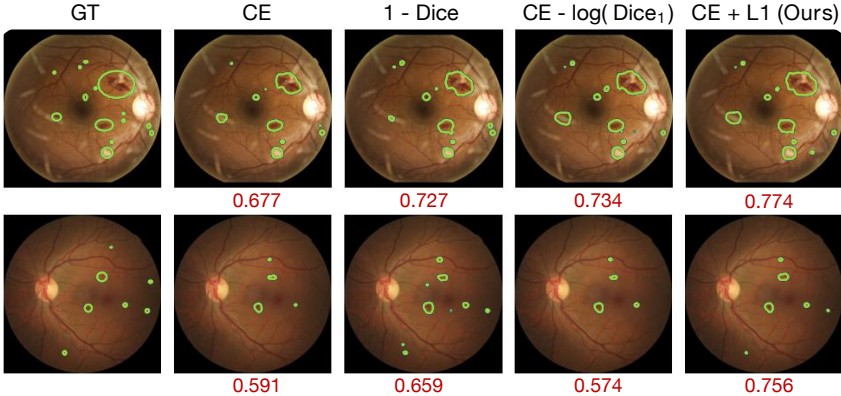

Figure 3: **Visual results on the Retinal Lesions dataset.** The ground-truth is depicted in the first column. At the bottom of each image, we indicate the corresponding obtained DSC score.

**Qualitative results on the Retinal Lesions dataset.** As shown in Fig. 3, we can observe that the model trained with Dice variants (*third* and *fourth column*) tends to undersegment large and medium target regions, while ignoring several small lesions and generating false positives. In contrast, the proposed solution enables a better trade-off between finding small regions, reducing the number of false positives and matching the size of the larger targets. Furthermore, one can notice that our loss yields better consistencies with the target region proportions than CE.

Table 3: **Results on Cityscapes validation set.** (Top: *Res50-FPN*, Bottom: *Res101-FPN*) The second row indicates the average region proportion (mRegProp) for each class. mIoU denotes the mean IoU score over all classes.

| | road | swalk | build. | wall | fence | pole | tlight | sign | veg. | terrain | sky | person | rider | car | truck | bus | train | mbike | bike | **mIoU** |
|---|---|---|---|---|---|---|---|---|---|---|---|---|---|---|---|---|---|---|---|---|
| mRegProp (%) | 32.6 | 5.4 | 20.2 | 0.6 | 0.8 | 1.1 | 0.2 | 0.5 | 14.1 | 1.0 | 3.6 | 1.1 | 0.1 | 6.2 | 0.2 | 0.2 | 0.2 | 0.1 | 0.4 | |
| CE | 97.2 | **80.6** | 90.0 | 38.2 | 51.3 | 55.7 | 60.5 | 72.2 | **91.0** | 58.6 | 92.3 | 75.6 | 48.5 | 92.5 | 50.5 | 67.7 | 56.3 | 50.2 | 71.6 | 68.45 |
| Focal loss | 97.1 | 78.8 | 89.5 | 37.9 | 48.9 | 51.8 | 56.9 | 67.6 | 90.6 | 56.9 | 92.6 | 73.1 | 42.8 | 92.0 | 47.5 | 57.5 | 46.1 | 49.2 | 68.4 | 65.53 |
| $1-$Dice | 95.9 | 77.6 | 87.7 | **41.1** | 47.6 | 56.0 | **66.2** | **74.8** | 90.0 | 59.6 | 93.1 | **77.3** | **55.8** | 91.1 | 10.0 | 63.9 | 34.6 | 53.6 | **73.6** | 65.77 |
| $-\log$ Dice | 94.0 | 70.8 | 85.6 | 33.2 | 39.1 | 50.6 | 62.2 | 69.6 | 88.4 | 52.1 | 88.1 | 74.2 | 50.7 | 89.2 | 34.8 | 55.3 | 31.7 | 48.0 | 70.7 | 62.54 |
| $CE-\log$(Dice) | 96.9 | 78.6 | 89.8 | 40.4 | 48.3 | **57.1** | 65.4 | 74.6 | 90.7 | 57.2 | 92.5 | 77.0 | 52.7 | 92.4 | 47.9 | 65.0 | 50.6 | 51.4 | 72.6 | 68.48 |
| $CE+\mathcal{D}_{KL}$(Ours) | 97.2 | 79.6 | 90.0 | 39.6 | 51.2 | 54.8 | 60.3 | 71.5 | **91.0** | 59.7 | 92.9 | 75.2 | 49.0 | 92.7 | **55.6** | 72.5 | 65.6 | 51.3 | 70.9 | 69.52 |
| **$CE+\mathcal{L}_1$(Ours)** | **97.4** | 80.2 | **90.1** | 37.9 | **51.9** | 55.7 | 61.0 | 72.1 | **91.0** | 58.9 | **93.5** | 75.7 | 49.8 | **92.8** | 54.9 | 70.2 | 62.8 | **54.0** | 71.6 | **69.55** |
| CE | **97.8** | 82.4 | 91.0 | 46.5 | 55.3 | 57.2 | 62.6 | 72.3 | 91.5 | 60.7 | **94.2** | 76.1 | 50.7 | 93.6 | 68.4 | 77.2 | 64.8 | 54.9 | 72.6 | 72.10 |
| Focal loss | 97.7 | 81.9 | 90.8 | **48.2** | 54.1 | 54.4 | 59.0 | 69.4 | 91.0 | 60.2 | 93.7 | 74.7 | 47.8 | 93.0 | 59.3 | 67.3 | 51.9 | 52.1 | 70.5 | 69.32 |
| $1-$Dice | 96.4 | 79.6 | 88.1 | 0.0 | 50.6 | **58.4** | **69.0** | **76.0** | 90.2 | 60.4 | 93.7 | **78.7** | **60.0** | 91.4 | 35.1 | 0.0 | 26.8 | 0.0 | **74.5** | 59.42 |
| $-\log$ Dice | 95.2 | 73.9 | 86.4 | 31.5 | 39.2 | 52.4 | 63.3 | 70.4 | 88.6 | 53.0 | 92.2 | 74.6 | 52.1 | 89.4 | 37.1 | 57.3 | 31.1 | 42.7 | 71.5 | 63.26 |
| $CE-\log$(Dice) | 97.2 | 79.9 | 90.2 | 42.5 | 52.2 | 57.6 | 66.7 | 74.8 | 90.9 | 59.6 | 93.7 | 77.7 | 57.4 | 92.9 | 56.4 | 72.1 | 58.9 | 54.2 | 74.0 | 71.00 |
| $CE+\mathcal{D}_{KL}$ (Ours) | **97.8** | 82.6 | 91.1 | 45.4 | **56.8** | 57.4 | 63.4 | 72.5 | 91.5 | **61.3** | 94.0 | 76.7 | 52.4 | 93.7 | 69.2 | 78.3 | 64.5 | 57.1 | 72.8 | 72.56 |
| **$CE+\mathcal{L}_1$ (Ours)** | **97.8** | **82.9** | **91.2** | 48.0 | **56.8** | 57.7 | 63.8 | 72.7 | **91.6** | 61.2 | 93.8 | 76.9 | 52.8 | **93.8** | **77.1** | **80.0** | **67.1** | **57.2** | 73.1 | **73.44** |

**Results on Cityscapes.** Table 3 reports the comparative per-class IoU and mean IoU (mIoU) on the validation set of Cityscapes with two network architectures. First, we can observe that in this multi-class dataset, regardless of the network, the proposed learning objectives outperform all the evaluated losses in terms of mIoU. Then, by investigating the relationship between mRegProp (second row in Table 3) and the corresponding segmentation performance across small region classes, we can observe that Dice-related losses have a hidden label-marginal bias towards extremely imbalanced solutions, preferring small structures. In particular, the linear Dice often obtains the highest mIoU for the smallest structures. This bias comes at the cost of less flexibility when dealing with arbitrary class proportions, which is reflected in its poor average mIoU (*right column*).

Quantitative evaluation on the Cityscapes test set is reported in Table 4. The observations in this table are consistent with the results on the validation set. Similarly to the validation set, our method achieves better results on both architectures. For the per-class scores on the test set, please refer to Appendix F.

It is noteworthy to highlight that both linear Dice and logarithmic Dice perform relatively poorly on Cityscapes, on average, empirically showing why Dice is rarely adopted in natural-image segmentation tasks. As we mentioned earlier, this might be due to its inherent label-marginal bias, which is inappropriate for segmenting regions with arbitrary class proportions. Furthermore, our simple solution yielded an improvement over CE by nearly 1.5%, regardless of the backbone. These empirical observations suggest that the proposed formulation in Eq. (5) results in better region-proportion guidance and training stability than existing segmentation losses. Finally, we can observe that employing $\mathcal{L}_1$ as regularizer consistently results in better performance than using the $\mathcal{D}_{KL}$ term. Hence, $\mathcal{L}_1$ might be a better option in extremely imbalanced segmentations, due to its gradient properties and stability when the label-marginal probability is close to 0.

Table 4: **mIoU on Cityscapes test set.**

| Loss | Res50-FPN | Res101-FPN |
|---|---|---|
| CE | 66.97 | 69.57 |
| Focal loss | 64.50 | 66.79 |
| $-\log$(Dice) | 59.77 | 62.62 |
| $CE-\log$(Dice) | 67.08 | 69.58 |
| $CE+\mathcal{D}_{KL}$ (Ours) | 68.35 | 70.07 |
| **$CE+\mathcal{L}_1$ (Ours)** | **68.35** | **70.11** |

## 4 CONCLUSION

We provided a detailed theoretical analysis of the two most popular semantic segmentation losses, *i.e.*, Cross-entropy and Dice, which revealed non-obvious bounding relationships and hidden label-marginal biases, suggesting that CE is a better option in general. Then, we showed how both loss functions could be written within a common formulation, containing a ground-truth matching term and a label-marginal bias. The implicit bias in Dice prefers small regions, improving its performance in highly imbalanced conditions, as in medical-imaging applications. The bias hidden in CE encourages the ground-truth region proportion, which makes it a generally better option in complex scenarios with diverse class proportions. Furthermore, we proposed a principled solution, which enables to control the label-marginal bias via $\mathcal{L}_1$ and KL regularizers that encourage the target class proportions, while improving training stability. Our flexible formulation enables the minority classes to have better influence on training, without losing adaptability to medium-to-large regions. Extensive experiments on a natural and medical imaging datasets validate the theoretical analysis in this paper, as well as the effectiveness of the presented solution.

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

## A   PROOFS

### A.1   PROPOSITION 1

*Proof.* Let us write the label-marginal bias term in the logarithmic Dice as a vector-valued function of probability simplex vector $\mathbf{p}$:

$$g(\mathbf{p}) = \mathbf{1}_K^\top \log(\mathbf{p} + \mathbf{y}) \quad \text{for} \quad \mathbf{p} = (\hat{p}_k)_{1 \leq k \leq K} \in \Delta_K \tag{6}$$

where symbol $\top$ denotes transpose and $\mathbf{1}_K$ is the $K$-dimensional vector of ones. Function $g$ is concave because its Hessian is a negative semi-definite matrix: The Hessian of $g$ is a diagonal matrix whose diagonal elements are given by $-\frac{1}{\hat{p}_k^2}$ and, hence, are all non-positive. Therefore, using Jensen's inequality and the fact that $\mathbf{p}$ is within the simplex, we have the following lower bound on penalty $g$ in Eq. (6):

$$g(\mathbf{p}) = g\left(\sum_{k=1}^{K} \hat{p}_k \mathbf{e}_k\right) \geq \sum_{k=1}^{K} \hat{p}_k g(\mathbf{e}_k) \tag{7}$$

where $\mathbf{e}_k \in \{0,1\}^K$ denote the $k$-th vertex of the simplex: the $k$-th component of $\mathbf{e}_k$ is equal to 1 while the other components are all equal to 0. Now, recall the definition of simplex vector $\mathbf{t} = (\hat{t}_j)_{1 \leq j \leq K}$: $\hat{t}_j = 1$ when $\hat{y}_j = \max_{1 \leq k \leq K} \hat{y}_k$ and $\hat{t}_j = 0$ otherwise. Given this definition, one could easily verify the following fact:

$$g(\mathbf{e}_k) \geq g(\mathbf{t}) \quad \forall k \tag{8}$$

To see this, let $j$ denotes the integer verifying $\hat{y}_j = \max_{1 \leq k \leq K} \hat{y}_k$ and let $k \neq j$. Then, we have:

$$g(\mathbf{e}_k) - g(\mathbf{t}) = \log\left(1 + \frac{1}{\hat{y}_k}\right) - \log\left(1 + \frac{1}{\hat{y}_j}\right) \geq 0. \tag{9}$$

This is due to the fact that function $\log\left(1 + \frac{1}{x}\right)$ is monotonically decreasing in $[0,1]$ and $\hat{y}_k \leq \hat{y}_j$. Now, combining inequalities (7) and (8), and using the fact that $\sum_{k=1}^{K} \hat{p}_k = 1$, we obtain:

$$g(\mathbf{p}) \geq g(\mathbf{t}) \tag{10}$$

$\square$

### A.2   PROPOSITION 2

*Proof.* Considering a generative view of the prediction model, random variable $\mathcal{F}$ associated with the learned features is continuous, while the random variable describing the labels, *i.e.*, $\mathcal{K}$, takes its possible values in a finite set $\{1, \ldots, K\}$. Then, the marginal distribution of the labels could be empirically estimated by the GT proportion of each segmentation region (as listed in Table 1) (Kim et al., 2005) :

$$\mathbb{P}(\mathcal{K} = k) \approx \hat{y}_k = \frac{|\mathbf{\Omega}_k|}{|\mathbf{\Omega}|} \tag{11}$$

Also, we express the conditional entropy of the learned features as follows :

$$\mathcal{H}(\mathcal{F}|\mathcal{K}) = \sum_{k}^{K} \mathbb{P}(\mathcal{K} = k)\mathcal{H}(\mathcal{F}|\mathcal{K} = k)$$

$$\approx \frac{1}{|\mathbf{\Omega}|} \sum_{k}^{K} |\mathbf{\Omega}_k|\mathcal{H}(\mathcal{F}|\mathcal{K} = k) \tag{12}$$

with each $\mathcal{H}(\mathcal{F}|\mathcal{K} = k)$ given by :

$$\mathcal{H}(\mathcal{F}|\mathcal{K} = k) = -\int_{\mathbf{f}^\theta} \mathbb{P}(\mathbf{f}^\theta|\mathcal{K} = k) \log \mathbb{P}(\mathbf{f}^\theta|\mathcal{K} = k) d\mathbf{f}^\theta \tag{13}$$

Hereafter, for notation simplicity, we omit $\mathcal{K}$ and use $\mathcal{H}(\mathcal{F}|k)$ instead of $\mathcal{H}(\mathcal{F}|\mathcal{K} = k)$. Also, we use $\mathbb{P}(\mathbf{f}^\theta|k)$ instead of $\mathbb{P}(\mathbf{f}^\theta|\mathcal{K} = k)$.

To estimate the conditional entropy in Eq. (13), let us refer to the following well known Monte-Carlo estimation (Kearns et al., 1997; Tang et al., 2019) :

**Monte-Carlo estimation.** For any discrete set of points $\mathbf{S} \subset \mathbf{\Omega}$, any function g and any feature embedding $\mathbf{f}$, we have :

$$\int_{\mathbf{f}} g(\mathbf{f})\mathbb{P}(\mathbf{f}|\mathbf{S}) \approx \frac{1}{|\mathbf{S}|} \sum_{i \in \mathbf{S}} g(\mathbf{f}_i) \tag{14}$$

where $\mathbf{f}_i$ denotes a feature vector at point $i$, and $\mathbb{P}(\mathbf{f}|\mathbf{S})$ stands for the density of $\{\mathbf{f}_i, i \in \mathbf{S}\}$.

Therefore, applying Montre-Carlo to $\mathcal{H}(\mathcal{F}|k)$ in Eq. (13), we can re-write Eq. (12) as follows :

$$\mathcal{H}(\mathcal{F}|\mathcal{K}) \approx -\frac{1}{|\mathbf{\Omega}|} \sum_{k}^{K} \sum_{i \in \mathbf{\Omega}_k} \log(\mathbb{P}(\mathbf{f}_i^{\theta}|k)) \tag{15}$$

Furthermore, using Bayes rule $\mathbb{P}(\mathbf{f}_i^{\theta}|k) \propto \frac{p_{ik}}{\hat{p}_k}$, in addition to the fact that $\sum_{i \in \mathbf{\Omega}_k} \log(\hat{p}_k) = |\mathbf{\Omega}_k| \log(\hat{p}_k)$, we obtain :

$$\begin{aligned}
\mathcal{H}(\mathcal{F}|\mathcal{K}) &\approx -\frac{1}{|\mathbf{\Omega}|} \sum_{k}^{K} \sum_{i \in \mathbf{\Omega}_k} \log\left(\frac{p_{ik}}{\hat{p}_k}\right) \\
&= -\frac{1}{|\mathbf{\Omega}|} \sum_{k}^{K} \sum_{i \in \mathbf{\Omega}_k} \log(p_{ik}) + \frac{1}{|\mathbf{\Omega}|} \sum_{k}^{K} \sum_{i \in \mathbf{\Omega}_k} \log(\hat{p}_k) \\
&= \text{CE} + \frac{1}{|\mathbf{\Omega}|} \sum_{k}^{K} |\mathbf{\Omega}_k| \log(\hat{p}_k) \\
&= \text{CE} + \sum_{k}^{K} \hat{y}_k \log(\hat{p}_k)
\end{aligned} \tag{16}$$

Finally, due to the definition of the label-marginal KL divergence, we have :

$$\mathcal{D}_{\text{KL}}(\mathbf{y}||\mathbf{p}) = \sum_{k=1}^{K} \hat{y}_k \log\left(\frac{\hat{y}_k}{\hat{p}_k}\right) \overset{c}{=} -\sum_{k}^{K} \hat{y}_k \log(\hat{p}_k) \tag{17}$$

This yields :

$$\text{CE} \overset{c}{=} \mathcal{H}(\mathcal{F}|\mathcal{K}) + \mathcal{D}_{\text{KL}}(\mathbf{y}||\mathbf{p}) \tag{18}$$

In summary, we give an information-theoretic prospective of CE. The entropy term can be considered as a ground-truth matching term, while the label-marginal KL term avoids trivial solutions and encourages the proportions of the predicted segmentation regions to match the ground-truth proportions.

$\square$

## B  THE BINARY SEGMENTATION CASE

In the two-class (binary) segmentation case, Dice might be used for the foreground region only (Milletari et al., 2016). Similarly to the multi-class case discussed in the paper, a single Dice term also decomposes into a ground-truth matching term and label-marginal penalty, with the latter encouraging extremely imbalanced binary segmentations. For this specific case, the logarithmic Dice and CE could be written as summations over the foreground and background segmentation regions:

$$-\log(\text{Dice}_1) \overset{c}{=} \underbrace{-\log\left(\frac{1}{|\mathbf{\Omega}_1|} \sum_{i \in \mathbf{\Omega}_1} p_{i1}\right)}_{\text{Foreground matching: DF}_1} + \underbrace{\log\left(\sum_{i \in \mathbf{\Omega}} p_{i1} + |\mathbf{\Omega}_1|\right)}_{\text{Label-marginal bias: DB}_1} \tag{19}$$

$$\text{CE} = -\underbrace{\frac{1}{|\mathbf{\Omega}_1|}\sum_{i\in\mathbf{\Omega}_1}\log p_{i1}}_{\text{Foreground matching: CE}_1} - \underbrace{\frac{1}{|\mathbf{\Omega}_2|}\sum_{i\in\mathbf{\Omega}_2}\log(1-p_{i1})}_{\text{Background matching: CE}_2} \tag{20}$$

In Eq. (19), the term $\text{DB}_1$ can be expressed, up to an additive constant, as a function of the marginal probability of the foreground class ($k=1$) as follows:

$$\text{DB}_1 \overset{c}{=} \log(\hat{p}_1 + \hat{y}_1) \tag{21}$$

Clearly, the marginal probability $\hat{p}_1$ measures the predicted proportion of pixels within the foreground region. This term reaches its minimum when the foreground region is empty ($p_{i1} = 0\,\forall i$). Therefore, since $\log$ is monotonically increasing, minimizing term $\text{DB}_1$ in Eq. (19) introduces a bias preferring small foreground structures. Note that this label-marginal term in the logarithmic Dice loss is important to avoid trivial solutions: when using the foreground-matching term alone, the model may assign all the pixels in the image to the foreground region.

The foreground-matching terms, $\text{CE}_1$ and $\text{DF}_1$, are closely related, with the former being and upper bound on the latter, due to Jensen's inequality: $\text{DF}_1 \le \text{CE}_1$. Both foreground-matching terms are monotonically decreasing functions of each softmax and reach their global minimum when all the softmax predictions in the ground-truth foreground are equal to 1 (*i.e.*, reach their target). Hence, the matching terms in Dice and CE can be viewed as two different penalty functions for imposing the same equality constraints, $p_{i1} = 1, \forall i \in \mathbf{\Omega}_1$, thereby encouraging the predicted foreground to include the ground-truth foreground.

## C    DISTRIBUTION OF REGION PROPORTIONS ACROSS CLASSES

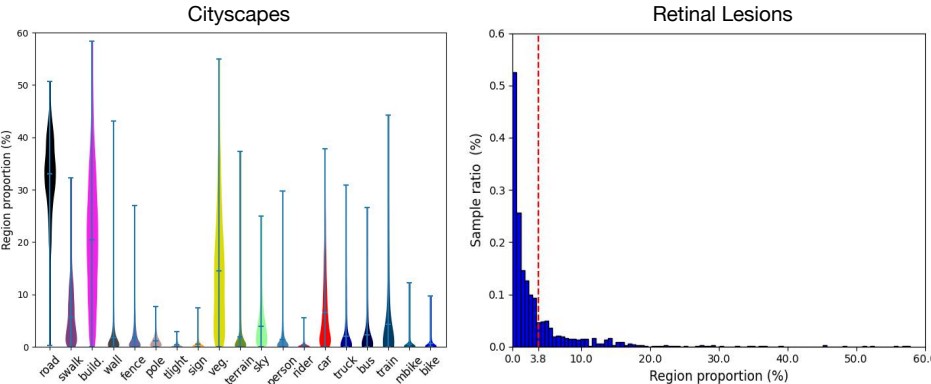

Figure C.1: **The distribution of region proportions for the** 19 **classes in Cityscapes dataset (*Left*) and Retinal Lesions dataset (*Right*).** The dashed line in the right figure indicates the average region proportion for Retinal Lesions. Best seen in color.

In Fig. C.1, we present the distribution of region portions for each class in Cityscapes and Retinal Lesions. Cityscapes has highly variable region sizes (ranging from an average of $1.0\%$ for rider and motorbike to an average of $32.6\%$ for road). In Retinal Lesions, although the average region proportion is $3.8\%$, there are some examples with large region proportions (above $50.0\%$), and the standard deviation is $6.6\%$. Thus, these plots present the significant heterogeneity on different classes found across these two data sets.

## D    THE TEMPERATURE SCALING

In our implementation, we employ a modified soft-max function with a temperature scaling parameter when computing the predicted region proportion $\hat{p}_k$ (also referred to as the predicted label-marginal probability, as in Table 1) :

$$s(\mathbf{z})_i = \frac{e^{\tau\cdot z_i}}{\sum_j^K e^{\tau\cdot z_j}} \tag{22}$$

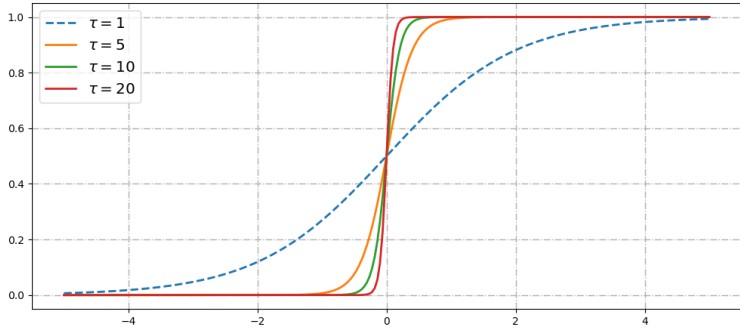

Figure D.1: **Comparison of soft-max functions with different temperature scaling paramter $\tau$.** This shows that the output confidence increases with larger $\tau$. Best seen in color.

where $\mathbf{z} = (z_i)_{1 \leq i \leq K}$ is the input vector of the soft-max function, and $\tau > 0$ acts as the temperature hyper-parameter. High values of $\tau > 0$ yield high confidence of the soft-max prediction, as shown in Fig. D.1: They push the softmax vector towards the vertices of the simplex, with prediction values approaching either $0$ or $1$. As a result, this enables a better estimate of the actual region proportion (or relative size). Throughout all our experiments, we set $\tau$ to $10$.

# E STUDY OF THE TRADE-OFF WEIGHT IN THE COMPOSITE LOSSES

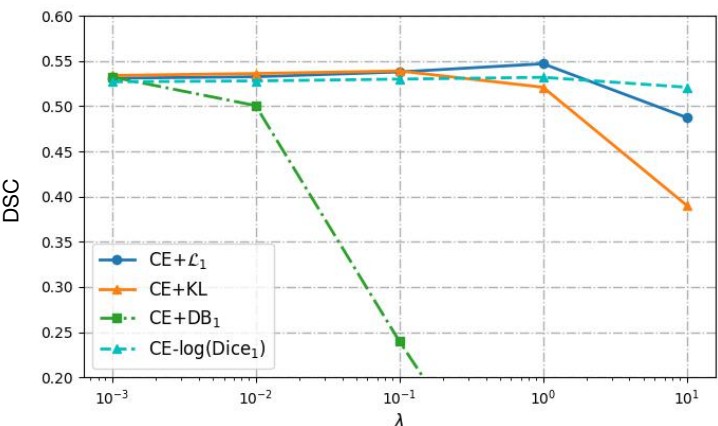

Figure E.1: The performances of different composite losses on the test set of Retinal Lesions with different values of the trade-off hyper-parameter $\lambda$. The network we used is fixed to Res50-FPN.

The proposed method is a type of composite loss. Thus, the trade-off hyper-parameter $\lambda$ could influence the final performance. Fig. E.1 shows the performances of the different composite losses examined in the paper, with different trade-off hyper-parameter values. While we plot the performances on the test set of Retinal Lesions, the behaviour on the validation set is similar, with the best trade-off parameters reached at the same value for both the validation and test sets.

From the curves, we can notice that $\mathcal{L}_1$ is a better choice than KL divergence for the proposed method because of its better stability. This may relate to its gradient properties and stability at the vicinity of $0$ (see Fig. 2 in the main document). Thus, we can use a relatively larger weighting values for CE+$\mathcal{L}_1$ (best performance achieved when $\lambda = 1.0$). Observe that CE+$\mathcal{D}_{KL}$ leads to comparable results at its best when $\lambda = 0.1$, but it seems more sensitive to the trade-off hyper-parameter. For the loss integrating CE with the Dice bias term, *i.e.*, CE+DB$_1$, we demonstrate that it can achieve performances similar to CE-log(Dice), but it drops significantly with high weighting values ($\lambda \geq 0.1$). This might be due to its gradient characteristics. Comparing to the widely suggested composite loss of CE and Dice, our method deliver better performance with the same hyper-parmeter budget.

## F  PER-CLASS RESULTS ON CITYSCAPES TEST SET

Table F.1: **Per-class Results on Cityscapes test set.** (Top: *Res50-FPN*, Bottom: *Res101-FPN*) The second row indicates the average region proportion (mRegProp) for each class. mIoU denotes the mean IoU score over all classes.

| | road | swalk | build. | wall | fence | pole | tlight | sign | veg. | terrain | sky | person | rider | car | truck | bus | train | mbike | bike | **mIoU** |
|---|---|---|---|---|---|---|---|---|---|---|---|---|---|---|---|---|---|---|---|---|
| mRegProp (%) | 32.6 | 5.4 | 20.2 | 0.6 | 0.8 | 1.1 | 0.2 | 0.5 | 14.1 | 1.0 | 3.6 | 1.1 | 0.1 | 6.2 | 0.2 | 0.2 | 0.2 | 0.1 | 0.4 | |
| CE | 97.5 | 79.3 | 90.3 | 45.0 | **46.3** | 53.5 | 61.6 | 67.3 | **92.1** | 70.1 | 93.5 | 78.4 | 53.3 | **93.7** | 41.4 | 48.8 | 38.2 | 55.2 | 67.1 | 66.97 |
| Focal loss | **97.6** | 79.1 | 89.8 | 41.3 | 43.6 | 49.9 | 57.7 | 63.8 | 91.7 | 68.4 | 94.1 | 75.7 | 47.3 | 93.0 | 37.8 | 44.9 | 35.6 | 50.5 | 63.8 | 64.50 |
| CE − log(Dice) | 97.4 | 79.2 | 90.3 | 43.8 | 45.0 | **55.1** | **67.0** | **70.3** | 91.8 | 70.0 | 94.0 | **79.4** | **56.0** | 93.3 | 39.7 | 50.8 | 32.2 | 51.2 | **68.1** | 67.08 |
| CE + KL (Ours) | **97.6** | **79.6** | **90.4** | **45.2** | 45.0 | 53.1 | 61.3 | 66.9 | 92.0 | **70.5** | 94.1 | 78.5 | 54.7 | 93.6 | **48.8** | **59.1** | 45.8 | **55.5** | 67.0 | **68.35** |
| **CE + $\mathcal{L}_1$ (Ours)** | **97.6** | 79.4 | **90.4** | 44.1 | 45.8 | 53.7 | 61.3 | 67.2 | **92.1** | 70.2 | **94.3** | 78.8 | 54.4 | **93.7** | 47.8 | 58.4 | **46.8** | 55.4 | 67.3 | **68.35** |
| CE | 97.8 | 81.3 | 90.9 | 45.6 | 50.2 | 55.7 | 64.2 | 67.1 | 92.3 | **70.9** | **94.6** | 79.7 | 57.6 | 94.2 | 51.7 | 58.0 | 43.5 | 58.1 | 68.5 | 69.57 |
| Focal loss | 97.9 | 81.3 | 90.6 | 46.4 | 46.9 | 52.4 | 59.9 | 65.7 | 91.9 | 69.4 | 94.4 | 77.3 | 53.0 | 93.6 | 42.7 | 48.7 | 35.9 | 54.5 | 66.5 | 66.79 |
| CE − log(Dice) | 97.7 | 80.0 | 90.5 | 43.8 | 48.5 | **55.8** | **68.1** | **70.9** | 91.7 | 70.5 | 94.2 | **80.6** | **61.4** | 93.7 | 46.9 | 57.6 | 40.6 | **59.2** | **70.4** | 69.58 |
| CE + KL (Ours) | 97.9 | 81.4 | 91.2 | 48.7 | 49.7 | 55.5 | 63.7 | 68.6 | **92.4** | 70.6 | **94.6** | 79.9 | 57.1 | **94.3** | **54.5** | **61.1** | 43.5 | 58.3 | 68.4 | 70.07 |
| **CE + $\mathcal{L}_1$ (Ours)** | **98.0** | **81.9** | **91.2** | **48.7** | **50.3** | 55.7 | 63.8 | 68.3 | 92.3 | 70.7 | 94.5 | 79.8 | 57.5 | 94.2 | 52.9 | 60.3 | **44.0** | **59.2** | 68.7 | **70.11** |

In Table F.1, we report the detailed per-class results on the Cityscapes test set. The proposed methods outperform related losses on both network settings. In term of mIoU, we achieve $68.35\%$ and $70.11\%$ on Res50-FPN and Res101-FPN, respectively. Compared to the baseline CE, our method yields improvement for most of the minority classes, like fence, train and motorbike, while it shows better adaptability for diverse classes than the Dice loss.

## G  MORE QUALITATIVE RESULTS ON THE RETINAL LESIONS DATASET

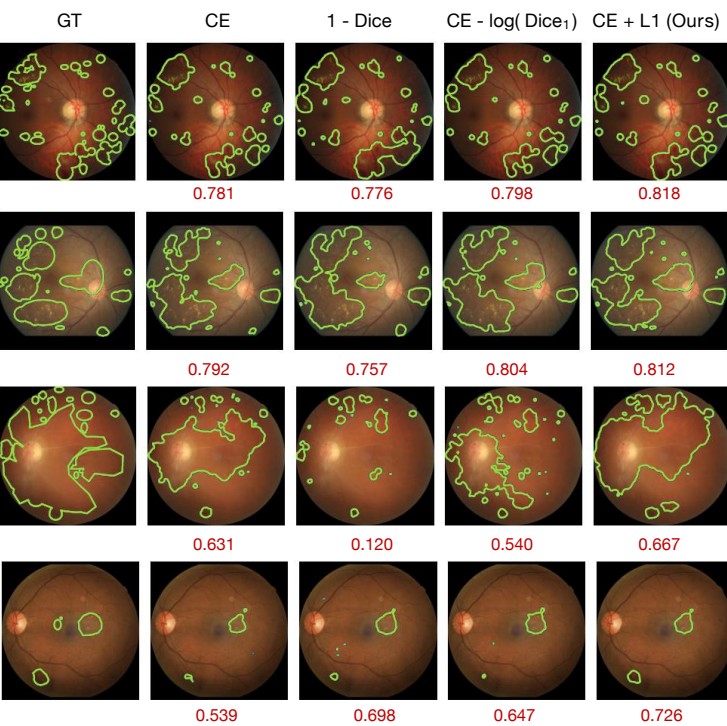

Figure G.1: **Additional visual results on the Retinal Lesions dataset.** The ground-truth is depicted in the first column. At the bottom of each image, we indicate the corresponding obtained DSC score.

Fig. G.1 gives more qualitative examples from Retinal Lesions Dataset. It is shown that our method is able to adapt to the highly variant segmenting regions from small to large, while Dice loss obviously degrades in the case of large regions like the example in the third row.

