# OpenReview forum: "The hidden label-marginal biases of segmentation losses"
_ICLR.cc/2022/Conference — ICLR 2022 Submitted_

### Official Review · Reviewer_njoP · 2021-10-20

**Correctness:** 3
**Technical Novelty And Significance:** 2
**Empirical Novelty And Significance:** 2
**Recommendation:** 5
**Confidence:** 3

**Main Review:**

Strengths:
1. This paper provides theoretical perspective about why Dice loss is better for smaller structure and CE loss is better at matching prediction distribution to training data distribution. While both phenomena are straightforward intuitively, a theoretical perspective from label marginal bias is appreciated.
2. This paper proposes a novel loss function that uses CE loss with a regularization term to correct label proportions.
3. Experiments on both natural and medical image segmentation problems with imbalanced classes were conducted to show that the proposed loss achieves better accuracy than Dice or CE loss themselves.

Weakness:
1. The proposed loss regularization term needs a hyper-parameter \lambda to control the weight of regularization. It is understandable that the optimal \lambda value is dataset-dependent due to different class distributions in different datasets. However, whether such \lambda is also network-dependent is not clear.

In this paper, only ResNet-FPN and ResNet-Unet are used in the experiment. More extensive experiment on different architectures using same lambda values may be needed. Or theoretical argument about why \lambda is only dataset-dependent and not network-dependent would be very helpful.

2. A useful baseline to compare to is "CE+w * Dice", where w is a weighting coefficient. As the proposed loss is "CE+\lambda * L1", the "CE+w * Dice" can provide probably similar performance with the same amount of hyper-parameter to tune.

3. Only two datasets, Cityscape and retinal lesions, are presented. It would be great if more public datasets with different class distribution characteristics can be used, so that we can gain more insights about how to tune \lambda to match datasets with different class distributions.

4. I agree with other reviewers on that the "segmentation GT" of the retinal lesion dataset is so poor that it cannot even be called segmentation annotation, but rather just detection annotation. In fact, without proper medical knowledge, I would say CE in Figure 3 is returning a better segmentation as they differentiate "normal" from "not-so-normal" more closely. Thus, higher DSC on this dataset is likely due to segmentation shape being smoother/more elliptical.

5. The Dice-related losses ("1-Dice" and "-log Dice") on **validation set** were highlighted by authors in Table 3, with extensive discussions in the first paragraph on page 9. However, Dice losses ("1-Dice" and "-log Dice") on **test set** were excluded in Table 4, without any discussion. I am not sure whether last paragraph of Sec.3 was for validation or test set. But either way, I think including a detailed comparison of Dice-related losses on test set is important, as the whole paper is essentially studying Dice loss and CE loss.


**Summary Of The Paper:**

This paper proposed theoretical analysis showing Dice loss has a bias preferring small structures, while losing flexibility of dealing with arbitrary class proportions. On the hand hand, the cross-entropy loss encourages proportions of the predicted segmentation to match the ground-truth proportions. A new loss function to control the label-marginal biases was proposed as sum of CE and weighted L1/KL divergence regularization.

**Summary Of The Review:**

The theoretical perspectives states the intuitively well-understood fact that Dice loss is better at capturing small objects while CE provides prediction distribution that matches training data. Thus, although it is an alternative perspective from label marginal bias, it did not lead to significant new knowledge about both losses. The proposed new loss, although achieving better accuracy on two datasets with two different networks, is not sufficiently validated in terms of how \lambda choice should be made regarding to different datasets and network structures. The comparison against "CE + w*Dice" is also strongly recommended, both theoretically and experimentally. The lack of Dice-related losses results on Cityscape dataset test set needs to be addressed. The poor quality of GT makes the retinal lesion dataset inappropriate to conduct segmentation experiments on.

---

> ### Author Response · Authors · 2021-11-23
> **Response to Reviewer njoP (PART 1/2)**
>
> We greatly appreciate the positive feedback and constructive comments. Nevertheless, we kindly disagree that the main message conveyed by our Propositions 1 & 2 (i.e., Dice and CE are closely related and essentially differ by their label-marginal bias) is intuitive and common knowledge within the community. As evidence that this is not the case: It is widely argued within the medical imaging community that Dice and CE are complementary losses, which motivated the use of compound CE-Dice losses; please refer, for instance, to the recent comprehensive review and experiments in [1]. Furthermore, this contrasts with the concerns from Reviewer QsrX, who found unclear why Dice loss favors extreme class imbalance. Also, were the messages in our Propositions 1 & 2 obvious, the highly competitive CE correction that we propose should have been already in use within the community (but this does not seem to be the case). Based on these arguments, we believe that the messages conveyed by Propositions 1 & 2 are neither intuitive nor obvious to show technically (details in the appendix).
>
> **1. About the balancing weight $\lambda$**
>
> > $\lambda$  is dataset-dependent due to different class distributions. Whether such $\lambda$ is also network-dependent is not clear.
>
> We would like to stress that the use of a balancing hyper-parameter $\lambda$ is not specific for our method, but for any other composite loss with more than a single term in the learning objective. In our experiments, we tuned $\lambda$ on the validation sets for all the composite losses (not only ours, but the existing losses such as CE-log(Dice)), for fair comparisons. The results are consistent across datasets and networks, and we found that the optimal value of $\lambda$ is quite independent from the dataset and network. We added clarifications in the experiments, sub-section "On the balancing weight in the composite losses".
>
> > "CE+w * Dice" can provide probably similar performance with the same amount of hyper-parameter to tune.
>
> In fact, we tuned w in compound loss CE-w.log (Dice) on validation data, for a fair comparison between all compound losses.
> In our experiments in Tables 2; 3; and 4, we also tuned the $w$ for CE- w.log(Dice) and reported the best scores. In the writing, we referred to this compound loss as CE-log(Dice) just to simplify notation. We found that fixing the $w$ to 0.1 yields consistent results. In the revision, we added a detailed ablation study in Sec. E of the appendix, showing that our method outperforms this composite loss with the same hyper-parameter budget. Please note that the CE - w.log (Dice) variant is often preferred in the literature over CE - w.Dice [2].
>
> **2. More experiments**
>
> > More extensive experiment on different architectures using same lambda values may be needed.
>
> Thanks for the suggestion. In the revision, we added new results for Res34-FPN and Res34-Unet on the Retinal Lesions Dataset (see Table 2):
>
> |              | Res34-FPN  |              | Res34-Unet |              |
> |:--------------|:------------:|:--------------:|:------------:|:--------------:|
> |      | DSC (%)    | HD-95 (mm)   | DSC (%)    | HD-95 (mm)   |
> | CE           | 51.4 (0.1) | 85.50 (2.04) | 52.4 (0.5) | 85.46 (3.71) |
> | Focal loss   | 51.2 (0.2) | 84.38 (4.86) | 51.2 (0.8) | 88.56 (4.90) |
> | 1-Dice       | 52.0 (0.7) | 84.80 (3.33) | 52.3 (0.7) | 89.19 (2.44) |
> | -log(Dice)   | 51.7 (0.9) | 86.02 (4.44) | 52.7 (0.5) | 85.84 (0.18) |
> | GDice        | 51.9 (0.7) | 86.70 (2.72) | 52.4 (0.5) | 89.99 (1.28) |
> | CE-log(Dice) | 52.2 (0.5) | 81.60 (6.70) | 52.6 (0.4) | 84.13 (2.40) |
> | CE+DB        | 51.7 (0.9) | 82.72 (1.66) | 51.8 (0.2) | 86.07 (1.83) |
> | CE+KL        | 52.7 (0.3) | 83.31 (2.40) | 52.8 (0.3) | 82.66 (2.40) |
> | CE+L1        | 52.8 (0.3) | 80.68 (2.49) | 53.1 (0.2) | 81.15 (2.75) |
>
> The conclusions are consistent with all the previous results (i.e., our method outperforms all related works in both metrics).
> Note that we report the scores by fixing the $\lambda$ across the networks and datasets.
>
> > Dice losses (”1-Dice” and ”-log Dice”) on test set were excluded in Table 4, without any discussion.
>
> We have included the test scores of -log(Dice) in Table 4 (59.77 for Res50-FPN and 62.62 for Res101-FPN). Thanks for this suggestion, which further demonstrates that Dice variants perform poorly on natural images, while our method can adapt to class-distribution variations. Please note that we only managed to add the test results of -log(Dice), due to the submission policy of Cityscapes test server (only available to submit once within 48h). Since the trend holds for both validation and test sets (see Table 3;4), this would not affect our conclusion, as both CE and Ours outperform 1-Dice by a large margin on the validation set.
>
> **References**
>
> [1] Ma et al., MedIA 2021. Loss odyssey in medical image segmentation.
>
> [2] Wong et al., MICCAI 2018. 3D Segmentation with Exponential Logarithmic Loss for Highly Unbalanced Object Sizes.

---

> > ### Author Response · Authors · 2021-11-23
> > **Response to Reviewer njoP (PART 2/2)**
> >
> > **3. Only two datasets, Cityscape and retinal lesions, are presented.**
> >
> > While we agree that adding more datasets is always welcome, we believe that the current experiments, including two completely different data sets, demonstrate the generalizability of the proposed method. Furthermore, similarly to our work, empirical validation on related works in this conference, i.e., works focusing on semantic segmentation, has been conducted over two datasets, e.g., [3, 4, 5, 6]. Therefore, we believe that using two different data sets only does not alter the main messages of the paper. We agree though that additional data sets will strengthen the paper, and plan to do this in future versions of the work.
> >
> > **4. About the quality of the Retinal Lesions Dataset.**
> >
> > > I agree with other reviewers on that the ”segmentation GT” of the retinal lesion dataset is so poor that it cannot even be called segmentation annotation, but rather just detection annotation.
> >
> > In fact, the statement by reviewer 3tvn that the retinal lesion segmentations are elliptical is incorrect, and dismisses the hard work of a panel of 45 experts in retinal images [7], who annotated about 35k lesions on over 1.5k retinal images from a widely used Kaggle retinal image dataset, and each image was assigned to at least three annotators according to [7]. To clarify this and address the concerns as to the visual and subjective quality of the ground truth, we have included additional images in the appendix of the revised version, where the ground-truth segmentations are not elliptical (please refer to Sec. G). We should kindly point out here that it is unusual (and should not be allowed by the conference) that reviewer 3tvn is releasing  the review publicly, and not in the appropriate official review section, thereby biasing the evaluations of  other reviewers.
> >
> > **References**
> >
> > [3] Ke et al., ICLR 2020. Universal weakly supervised segmentation by pixel-to-segment contrastive learning.
> >
> > [4] Guizilini et al., ICLR 2019. Semantically-guided repre-sentation learning for self-supervised monocular depth.
> >
> > [5] Casanova et al., ICLR 2019. Reinforced activelearning for image segmentation.
> >
> > [6] Zou et al., ICLR 2020. Pseudoseg: Designing pseudo labels for semantic segmentation.
> >
> > [7] Wei et al., ICPR 2020. Learn to segment retinal lesions and beyond.

---

> > > ### Comment · Reviewer_njoP · 2021-11-24
> > > **Response to Response to Reviewer njoP (PART 2/2)**
> > >
> > > Even though the newly added Sec.G in appendix showed that retinal lesion dataset GT contours are not entirely of ovals, large amount of segmentation GT contours are still made of compound ovals or polygons with relatively poor details in boundary precision. I still strongly believe that adding another dataset with annotation with better boundary precision would make the results significantly more convincing.
> > >
> > > Another question is that retinal lesion dataset include segmentation mask of eight different classes [A]. But this draft chooses to combine all eight classes into one foreground class in the experiment. Could the authors explain why?
> > >
> > > [A] Qijie Wei, Xirong Li, Weihong Yu, Xiao Zhang, Yongpeng Zhang, Bojie Hu, Bin Mo, Di Gong, Ning Chen, Dayong Ding, and Youxin Chen. Learn to segment retinal lesions and beyond. In ICPR, 2020.
> > >
> > > Additionally, I do not appreciate the emotionally charged replies, borderline attacks, given by authors. For example, the authors reply claims that my review "dismisses the hard work of a panel of 45 experts in retinal images". However, in the *original* draft, the authors did *NOT* mention how the ground truth segmentation was obtained in this dataset *at all*. Moreover, *only* elliptical GT contours were shown in Figure 3. With all the available information presented in the original draft, I raised a reasonable concern. I appreciate the more complete information provided by the revised draft and authors' reply regarding the dataset, and I stand corrected. However, such emotional statement is not necessary nor helpful in promoting a healthy technical discussion.

---

> > > > ### Author Response · Authors · 2021-11-29
> > > > **Follow-up answer to Reviewer njoP (Part 1)**
> > > >
> > > > Thank you for the prompt and detailed response and references. As stated in our initial answer, we greatly appreciate the comments of Reviewer njoP, which we found constructive, both in the initial review and in this detailed follow-up. We will fully account of these comments in the next versions of the paper. Here are our responses to the latest two sections of comments by Reviewer njoP.
> > > >
> > > > **1) About the emotionally charged answer**
> > > >
> > > > First of all, we apologize to Reviewer njoP if the reviewer felt our answer was "emotionally charged". This was not our intention at all. We concede that the statement "dismisses the hard work of a panel of 45 experts in retinal images" is a bit harsh in light of the information we provided in the initial draft (as we just cited the retinal dataset paper). However, we emphasize that, in our answer to Reviewer njoP in this retinal-dataset matter, we carefully said: "In fact, the statement by Reviewer 3tvn that the retinal lesion segmentations are elliptical is incorrect, and dismisses the hard work..." So, we clarify that our statement was solely intended to Reviewer 3tvn, in response to the overall misleading/non-informative/short review provided by 3tvn.
> > > >
> > > > **2) Retinal data set has poor details in boundary precision. I still strongly believe that adding another dataset with annotation with better boundary precision would make the results significantly more convincing.**
> > > >
> > > > As acknowledged in our initial answer, we agree that additional data sets could only make the message stronger, and we are currently working on this. Also, let us assume that the retinal data set has poor boundary precision. However, the paper has also comprehensive experiments on Cityscapes with high boundary precision, and we reached consistent conclusions from both the results of Cityscapes and Retinal Lesions. Finally, while we concede that the concern about the boundary precision in retinal data is legitimate, we kindly disagree that the level of boundary precision invalidates the conclusions reached on the retinal dataset (as the ground-truth is the same for all the competing losses and is built independently by experts).
> > > >
> > > > **3) Multi-class setting vs. binary setting in Retinal Lesions**
> > > >
> > > > In fact, we wanted to cover the binary case as, typically in this case, the Dice term is used for the foreground region alone
> > > > (not the background). In this case, the label-marginal bias has a different form (which we detailed in the appendix).
> > > > This specific binary form of Dice (i.e., only one Dice term for the foreground) appeared initially in the paper by [Milletari et al., 2016], which pioneered the use of the Dice loss and is, therefore, widely used within the community. Therefore, since we already have the multi-class case in the Cityscapes dataset, we wanted to also cover this widely used binary case of Dice. We could also include the multi-class case for the Retinal data set in the paper.
> > > >
> > > > [Milletari et al., 2016] Fausto Milletari, Nassir Navab, and Seyed-Ahmad Ahmadi.  V-net: Fully convolutional neural net-works for volumetric medical image segmentation. In3DV, 2016.

---

> > ### Comment · Reviewer_njoP · 2021-11-24
> > **Response to Response to Reviewer njoP (PART 1/2)**
> >
> > First of all, I would like to correct the authors' misunderstanding of my review. My review stated that the phenomena that "Dice loss is better for smaller structure and CE loss is better at matching prediction distribution to training data distribution" is well-known in the field, not that "Dice and CE differs on label-marginal bias" is well-known. My review acknowledged that "a theoretical perspective from label marginal bias is appreciated". I  hope this can help eliminate the authors' "surprise".
> >
> > Second, linear Dice instead of log(Dice) is as equally commonly, if not more commonly, used in medical imaging community than log(Dice). For example, in the recent comprehensive review ref[1] provided in the authors' reply, both types of Dice loss definition are linear form (not log form). In ref[1] Eq.(22), the Dice-CE combo loss is also in the form of L_CE + L_Dice instead of L_CE + log(L_Dice). The following references all uses linear Dice loss or CE+linear Dice losses:
> >
> > [A] Isensee, Fabian, et al. "nnU-Net: a self-configuring method for deep learning-based biomedical image segmentation." Nature methods 18.2 (2021): 203-211.
> > [B] Milletari, Fausto, Nassir Navab, and Seyed-Ahmad Ahmadi. "V-net: Fully convolutional neural networks for volumetric medical image segmentation." 2016 fourth international conference on 3D vision (3DV). IEEE, 2016.
> > [C] Alom, Md Zahangir, et al. "Recurrent residual U-Net for medical image segmentation." Journal of Medical Imaging 6.1 (2019): 014006.
> > [D] Bertels, Jeroen, et al. "Optimizing the dice score and jaccard index for medical image segmentation: Theory and practice." International Conference on Medical Image Computing and Computer-Assisted Intervention. Springer, Cham, 2019.
> >
> > Thus, it would be desirable to have the CE+linear Dice loss more carefully studied in the experiment, instead of only presenting log(Dice) or CE+log(Dice) performances. Because as mentioned by author in Sec.2.4: "this does not mean that optimizing these two variants (linear Dice vs. log(Dice)) leads to exactly the same results".
> >
> > Third, the concern surrounding how to set \lambda is only partially addressed. The statement "the optimal \lambda is quite independent from the dataset and network" will need stronger evidence to back up. The authors added the 34-layer variant of the original networks (Unet and FPN) in the revised draft. However, the only architectures presented remains being only ResNet-based Unet and FPN. Other state-of-art semantic segmentation networks, such as DeepLab and OCR, whose designs are significantly different from the presented networks, are not studied to show the robustness of lambda value setting. In terms of datasets, various medical image segmentation datasets showing different class distributions, were not studied. Also, original retinal lesion dataset contains segmentation of eight different classes. But only one-class segmentation results were shown in the experiment.
> >
> > Using the same optimal \lambda tuned on retinal lesion dataset on cityscape shows promise of a robust \lambda setting to some degree. But does the optimal \lambda tuned on cityscape works equally good on the retinal lesion dataset? In Figure E.1, CE+L1 seems to be marginally better than CE-log(Dice) only at certain \lambda values. This seems to indicate that \lambda need to be tuned in order to outperform the CE-log(Dice) baseline. How about comparison against CE+linear Dice baseline?

---

> > > ### Author Response · Authors · 2021-11-29
> > > **Follow-up answer to Reviewer njoP (Part 2)**
> > >
> > > **4) About the perspectives of Proposition 1 & 2 and clarification of our misunderstanding of njoP review**
> > >
> > > We appreciate the clarification and correction of our understanding of njoP review. Thanks! Perhaps, we mis-interpreted the word "well-understood" in the following statement by Reviewer njoP: "The theoretical perspectives states the intuitively well-understood fact that Dice loss is better at capturing small objects while CE provides prediction distribution that matches training data. Thus, although it is an alternative perspective from label marginal bias, it did not lead to significant new knowledge about both losses." To our knowledge, this fact about Dice being better for small regions is widely verified/known experimentally (as we stated in the paper) but is not "well-understood", neither theoretically nor intuitively. We said we are "surprised" because, for us, it did not seem intuitive at all (as Dice is a measure of overlap between ground-truth and predictions). By the way, similarly to us, Reviewer QsrX did not found the result obvious/intuitive. If Reviewer njoP has in mind an intuition or a simple explanation about why Dice prefers small regions (beyond the wide experimental evidence in the literature, which we acknowledged in the paper), or could point to a reference doing so, we will be very happy to learn about it. We also kindly disagree that our results in Props 1 and 2 do not lead to significant new knowledge: Our theoretical results challenge the wide use of Dice and point to the fact that CE with the proposed simple correction should be preferred over Dice in all cases. In our humble opinion, this is quite significant, given the wide use of Dice within the community (as stated by Reviewer njoP).
> > >
> > > **5) CE+linear Dice**
> > >
> > > We will include CE+linear Dice in the paper. Thanks for pointing to this! In fact, in our preliminary experiments, we found that the results of CE+linear Dice are marginally lower than CE+log Dice, so we didn't include them (we wanted to avoid overloading the Tables unnecessarily and to focus on confirming the main theoretical insights).
> > >
> > > **6) Further experiments on hyper-parameter lambda**
> > >
> > > We also agree that further experiments on the robustness of the choice of hyper-parameter lambda with respect to segmentation networks and data sets can only strengthen the experiments, as it will convey a desirable feature of the proposed compound loss. However, we did not fully understand why the fact that hyper-parameter lambda vary across networks and tasks is an issue in practice (as long as the best lambda is obtained over independent validation data sets, as is the case of all compound losses). In deep learning, in general, there is a large number of crucial hyper-parameters (learning rate, weight decay, early stopping, batch size, other optimization parameters, among many others), many of which vary across networks and data sets/tasks. The architecture itself could be viewed as a large set of hyper-parameters that change from one task to another.

---

> ### Comment · Reviewer_njoP · 2021-11-29
> **Answers to authors' questions. Original recommendation unchanged.**
>
> Thank the authors for providing additional feedback to my responses. Below are reply/clarification to the authors' responses, as well as my final recommendation.
>
> Intuition about why Dice is good at small regions:
>
> One intuitive perspective about why Dice is performing well at smaller regions is that its definition only depends on foreground (unlike CE, Dice loss is not computed on background class). Thus, no matter how small the foreground region is (even if foreground occupies only 1% of image), the loss encourages finding them. Otherwise, a very high penalty will be incurred.
>
> Hyper-parameter lambda:
>
> The proposed loss is essentially a compound loss of CE and L1 norm regularization weighted by lambda. This extra parameter introduces additional burden for the user to use it in practice. Thus, I am concerned about how sensitive/robust it is regarding to different networks and datasets.
>
> It is possible that a new loss with a tunable hyper-parameter may be suitable in only limited scenarios, and extensive experiments across sufficiently different types of datasets and networks are needed to either confirm or deny it. In fact, even if the new loss may have limitations, identifying these limitations can be very informative and further inspire new research directions.
>
> The current validation on retinal lesion dataset, which has relatively poor details in boundary precision, cannot support the authors' claim that "the optimal value of \lambda is quite independent from the dataset and network".
>
> In sum, the reviewer appreciates the theoretical perspectives provided on the CE and Dice loss from the label marginal bias perspective. However, the accordingly proposed new loss, which is CE+\lambda L1 norm regularization, introduces a hyper parameter that may depend on both datasets and network, with no easy intuitive rules or theoretical guidelines to set it. In terms of networks, only ResNet-FPN and ResNet-Unet with 34 and 50 layers networks were demonstrated, with no other SOTA networks. In terms of datasets, as the retinal lesion dataset's annotation has relatively poor details in boundary precision, the variety of datasets is not enough to either confirm or deny the robustness of the weighting hyper parameter \lambda of the L1 norm regularization, or provide enough insight about how to set it according to different datasets with different class distributions. After careful reviews of the authors' revised work, replies, and other reviewers' opinions, the above concerns remain not sufficiently addressed. Thus, I choose to keep my original decision unchanged: 5-marginally below the acceptance threshold.

---

### Official Review · Reviewer_QsrX · 2021-11-02

**Correctness:** 3
**Technical Novelty And Significance:** 3
**Empirical Novelty And Significance:** 3
**Recommendation:** 6
**Confidence:** 4

**Main Review:**

Strengths:
+ The question addressed in this work is inherently an interesting one. Studying the relation between two popular loss functions which are understood to be somehow complementary to each other and attempting to make connections between them can be an important contribution.
+ The decomposition of the loss functions into the ground truth penalty and label bias terms is reasonable, and the proofs provided in A1. and A2 are mostly adequate.
+ Experiments with the L1 regularization for the class imbalance dataset (retinal segmentation) show promising results.

Weaknesses:


- One of my major concerns is in the proposed strategies to modulate the weighting of the label bias term in CE. The key argument is that the relative contribution of the label-marginal term is of smaller magnitude in class imbalance cases. This is not obvious from the Propositions 2. How is this conclusion drawn?

- Two regularization strategies are proposed; one of which is KLD term D(y||p). But, in  Eq. 4 CE already decomposes into entropy and the same KLD term. Are these two KLD terms the same? If so, what is the influence of the additional regularization? Perhaps it only serves as a scaling of the label bias term. In that case, how does the L1 regularization influence the scaling of the label bias term.

- L1 regularization is posited to be better than KLD regularization, and one of the reasons cited is its "gradient properties and stability".  This is somewhat unconvincing. Is this in comparison with KLD?

- The specific choice of the two types of regularization terms is simply stated. Why were these two types of regularization schemes chosen? What was the motivation? How were these choices made? Is there a theoretical justification for using the L1 term?

 - The decomposition of the losses into two competing terms and their interpretations are reasonable. However, for Dice loss, the arguments presented for why the label marginal bias term in Eq. 1 end up favouring extreme class imbalance is unclear. Additional explanation can be helpful here.

Minor comments:
- An important point in the paper is the competing nature of the ground truth penalty and label bias term. Their competing nature should be elucidated further. It is not immediately obvious in case of the CE (Eq.4). and can be helpful to readers.

**Summary Of The Paper:**

This work presents a decomposition of two commonly used segmentation losses (cross entropy/CE and Dice losses) into two competing components. The two components are interpreted as ground truth matching and label-marginal penalties. This work argues that the label-marginal penalties in Dice loss favour extreme  class imbalances, and hence their widespread use in medical imaging segmentation tasks which encounter class imbalance more commonly. The label marginal terms in CE is argued to be better as this term matches the label marginals to the ground truth distributions. The scaling between the two terms in CE is presented as the problem and this work proposes an L1 regularization to overcome the label bias problem in CE. Experiments on two segmentation tasks show interesting influence of these regularizations.

**Summary Of The Review:**

The decomposition of the losses into two terms is interesting and largely rigorously studied. However, the regularization schemes proposed are not well justified.

---

> ### Author Response · Authors · 2021-11-23
> **Response to Reviewer QsrX (PART 1/2)**
>
> We thank the reviewer for the effort, for finding the contribution interesting/important, and greatly appreciate the constructive concerns raised to improve the paper.
>
> **1. About the proposed strategies to modulate the weighting of the label bias term in CE.**
>
> Thanks for this comment. We agree that this might need a few more sentences for clarification. In fact, for class imbalance, the difficulty in the ground-truth matching term in Eq. (3) and in CE is that, for each region, we have a summation over the region. This results into several-order-of-magnitude difference in the value of the regional terms in Eq. (3) and in CE, which we argue might affect the contribution of samples in the minority classes during gradient-based training (as their corresponding regional summation in the loss is significantly smaller). This is not the case in the label-marginal terms where the regional terms are within a much narrower scale variation  (due to L1 and logarithmic scale of region size in KL). Furthermore, our argument that the relative contribution of the label marginal term is essentially what makes the difference resonates with the fact that compound losses CE-Dice with an adjustable balancing weight are commonly considered as the state-of-the-art options in the medical imaging community[1]. Our theoretic analysis suggests that CE, with our label-marginal correction, should be preferred over widely used Dice in all cases, including imbalanced medical-imaging problems, as it promotes the right label-marginal distribution, rather than the non-informative bias term of Dice.
>
> **2. About the choices between KL and L1 in our method**
>
> >In Eq. 4, CE already decomposes into entropy and the same KL term. Are these two KL terms the same? If so, what is the influence of the additional regularization? Perhaps it only serves as a scaling of the label bias term. In that case, how does the L1 regularization influence the scaling of the label bias term.
>
> > L1 regularization is posited to be better than KL regularization, and one of the reasons cited is its "gradient properties and stability". This is somewhat unconvincing. Is this in comparison with KL?
>
> > The specific choice of the two types of regularization terms is simply stated. Why were these two types of regularization schemes chosen? What was the motivation? How were these choices made? Is there a theoretical justification for using the L1 term?
>
> Thank you for these comments. Yes, in principle, any divergence measure could be used as long as it promotes the right label-marginal distribution. KL is a natural choice as it appears in our Proposition 2. Indeed it is the same KL, with the only difference being the scaling factor. As discussed in answer to Q.1 above, the crucial difference for KL is the scaling factor: In CE, this factor is implicit (hidden) and uncontrollable. We argue that controlling the relative contribution of the bias term, as hinted by our analysis in Proposition 2, plays an important role for adapting to different scenarios and drive the attention of training towards matching the ground-truth proportions, regardless of the divergence employed. This also resonates with the experimental evidence in the literature that compound CE-Dice losses with an adjustable balancing weight are highly competitive.
>
> The main reason that we recommend using L1 lies in its more stable gradient dynamics, which we believe is desirable for gradient based optimization. As illustrated in Fig. 2, the derivative of KL w.r.t label-marginal probability is infinity at the vicinity of 0, where as L1 has a constant derivative w.r.t to the label-marginal probability near 0. This might be important in extreme class imbalance situations. This observation is also supported by our experiments showing that CE+L1 performs generally better than CE+KL (Table 2;3;4).
>
> **3. The competing nature of the ground truth penalty and label bias term is not immediately obvious in case of the CE (Eq.4)**
>
> In Eq. 4, we show that the CE loss is equivalent, up to an additive constant, to the sum of these two competing terms: ground-truth matching and label-marginal bias. These two terms have different objectives, and thus minimizing them results in competing goals: On the one hand, please observe that minimizing the entropy reaches its goal when all the pixels (within each region) are assigned to the same class, which might result in trivial solutions (all pixels in the image assigned to the same class). On the other hand, the second term reaches its minimum when the class proportions from the ground truth and predicted probabilities match each other, which avoid the trivial one-class solutions of the entropy term alone.
>
> **References**
>
> [1] Ma et al.,MedIA 2021.Loss odyssey in medical image segmentation.
>
> [2] Chen et al.,ECCV 2018.Encoder-decoder with atrous separable convolution for semantic image segmentation.
>
> [3] Yuan et al.,ECCV 2020.Object-contextual representations for semantic segmentation.

---

> > ### Author Response · Authors · 2021-11-23
> > **Response to Reviewer QsrX (PART 2/2)**
> >
> >
> > **4. For Dice loss, the arguments presented for why the label marginal bias term in Eq. 1 end up favouring extreme class imbalance is unclear.**
> >
> > In fact, the bound relationship in Proposition 1 and Eq. (2) shows that the global minimum of the label-marginal bias in Eq. (1) is reached at the vertex of the simplex, i.e.: A specific region includes all the pixels while all the remaining regions are empty. Therefore, its minimization favours extreme class imbalance.

---

> > > ### Comment · Reviewer_QsrX · 2021-11-29
> > > **Sufficiently satisfying response to reviews (Raising score from 6 to 7)**
> > >
> > > The authors' response to all reviewers is largely thorough and they have also addressed my major concerns to a satisfying extent. The elaborate discussions presented justifying the experiments on the retinal dataset, additional measures reported in the results table and clarifications presented in response to all reviews cumulatively improve my assessment of this work. As a result, I am willing to change the score from 6 to 7.
> > >
> > > Finally, I would like to assure the authors that my reviews were certainly not swayed by anything other than their work and responses presented here. The implication by the authors that the reviewers could have become biased due to some early comments is not fair.

---

> > > > ### Author Response · Authors · 2021-11-29
> > > > **Follow-up answer to QsrX**
> > > >
> > > > We greatly appreciate the informative, constructive and fully independent review by QsrX and by the other reviewers, and we
> > > > concede that our statement about the fact that the early comments of Reviewer 3tvn might have biased the other reviews is not totally fair.  Our apologies for that.

---

### Official Review · Reviewer_P7HM · 2021-11-02

**Correctness:** 3
**Technical Novelty And Significance:** 3
**Empirical Novelty And Significance:** 3
**Recommendation:** 6
**Confidence:** 3

**Main Review:**

Strengths
- The paper presents an interesting analysis for the CE and Dice loss function, indicating their relationships in terms of the called label-marginal bias. Base on this, the paper proposes an alternative variation of the CE loss function for segmentation.

Weaknesses
- Regarding the performance on the medical dataset, are the performance gains similar in other medical datasets, for example, organ segmentation (where some structures can be considerably smaller compared with the background, especially when considering 3D input data)? It is possible to discuss the applicability to different datasets, and in what cases, can the proposed function more/less appropiate

- Some points that might not be clear are the relation of the CE, DSC, and the label marginal concerning the performance of models in computer vision, and medical images. In this sense, it would be interesting to elaborate on this regard.

- Do the CE-based functions used in the experiments correspond to the weighted versions?

**Summary Of The Paper:**

The paper presents an analysis of the cross-entropy (CE) and dice loss (DSC) for segmentation tasks,  in terms of the label-marginal bias.  According to the analysis, DSC prefers small regions, while CE encourages a prediction that has a similar proportion to the ground truth. Considering this analysis, the work proposes an alternative loss function consisting of the cross-entropy augmented with a regularization term (L1 or KL divergence) that operates on the prediction of the network. Different experiments compare the presented loss function of previous proposals.

**Summary Of The Review:**

Segmentation loss functions are employed in different medical and real-world datasets, particularly the medical domain can present a wide variability in available datasets.  It might be interesting to discuss results on additional datasets, compared with the DSC (as the paper discussed, it is widely used in the medical domain).

---

> ### Author Response · Authors · 2021-11-23
> **Response to Reviewer P7HM**
>
> We thank the reviewer for the time spent on our paper, and appreciate the positive feedback as well as the the constructive concerns raised to improve our work, which mostly stem from unclear points.
>
> **1. Regarding the performance on the medical dataset, are the performance gains similar in other medical datasets.**
>
> Thanks for the suggestion of including more datasets in the experiments, which would definitely strengthen the experiments.
> We choose specifically the current two data sets in the experiments, one based on medical images and the other on natural images, so as to convey and validate the main theoretical insights about CE and Dice, as well as the effect of the proposed method. Indeed, one of the main insights is that, while DSC loss performs well on highly imbalance scenarios, it cannot properly handle datasets with heterogeneous class distributions within the data set. The choice of the retinal dataset was inspired by this fact, as the lesions in these images could be extremely small, particularly compared to the background, as suggested by the reviewer. The same, but to a lesser extent, applies to the computer vision dataset, Cityscapes, where the target sizes are highly variant. While we agree that adding more datasets is always welcome, we believe that the current experiments demonstrate the generalizability of the proposed method. We will add additional datasets in future versions of the work.
>
> **2. Some points that might not be clear are the relation of the CE, DSC, and the label marginal concerning the performance of models in computer vision and medical images.**
>
> Our Propositions 1 & 2 reveal that, the common belief and the widely used argument that CE and Dice are complementary does not hold, and that CE and Dice are, in fact, closely related with the main difference between them being the label-marginal bias .
>
> The main points are :
>
> - Both CE and DSC decompose into a common framework containing two penalty terms: a closely related ground-truth matching term and a label-marginal term preventing trivial solutions.
>
> - They essentially differ in the label-marginal bias encoded in the second term, i.e., CE encourages the right region proportions while Dice prefers predicting extremely small regions.
>
> The above observation perfectly explains a large experimental evidence in the literature, with Dice related losses performing very well in medical imaging applications [1] where there exists extreme class imbalance and the number of samples is usually limited. On the other hand, CE dominates natural images and computer vision applications [2, 3], as a result of the highly variant target sizes in natural datasets.
>
> Our method can be generally better under different types of scenarios because:
>
> - Compared to Dice, we always encourage the right region proportions instead of the non-informative bias, particularly in natural datasets.
>
> - Compared to CE, we provide an explicit solution to control the contribution of the label-marginal penalty.
>
> This is why we conducted the experiments on one medical imaging dataset and one natural-image dataset to validate the theoretic relation between CE, Dice and the proposed method. As shown in Table 2, on the Retinal Lesions dataset, the results of Dice-related losses, i.e., 1-Dice and -log(Dice), are comparable or slightly better than CE, depending on the backbone, whereas on the Cityscapes dataset (Table 3), CE outperforms Dice by a large margin. The proposed method performs better on different datasets and networks, showing its better generalizability.
>
> **3. Do the CE-based functions used in the experiments correspond to the weighted versions?**
>
> We used the standard non-weighted CE loss in the experiments. Though weighted CE (WCE) is appealing, it is empirically demonstrated by the comprehensive recent experimental study in [1] that WCE significantly under-performs other standard losses, including its non-weighted counterpart. We believe that WCE has over-fitting problems when the number of samples do not reflect the actual distribution, which is particularly common in medical imaging applications. This explains why we employed the original version as the baseline in our experiments.
>
> **References**
>
> [1] Ma et al., MedIA 2021. Loss odyssey in medical image segmentation.
>
> [2] Chen et al., ECCV 2018. Encoder-decoder with atrous separable convolution for semantic image segmentation.
>
> [3] Yuan et al., ECCV 2020. Object-contextual representations for semantic seg-mentation.

---

### Official Review · Reviewer_3tvn · 2021-11-04

**Correctness:** 4
**Technical Novelty And Significance:** 1
**Empirical Novelty And Significance:** 1
**Recommendation:** 3
**Confidence:** 5

**Details Of Ethics Concerns:**

- After providing an explicit bound relationship an information theoretic analysis, the paper proposes to add KL divergence and L1 norm to control explicitly the label- marginal bias. Moreover, the paper claim to address imbalanced data. It is not clear how to regularization term address that problem.
- Many recent loss functions for imbalanced data are not included in the paper.
- The grountruth on retinal lesions dataset is not reliable because the lesion is shown in an oval.
- There is no ablation study on imbanced data or training stabability.
- The proposed loss provides similar performance with other well known losses.

**Main Review:**

Strength: Good motivation, Good organization, Readable
Weakness: The paper analyzes Dice and CE losses while there are many recent losses ignored.
The proposed loss is mainly based on CE loss and adds one more regularization term.
The proposed loss is tested on Retinal Lesions dataset which groundtruth RoI is labeled as an oval which is not accurate for the segmentation task.
Compare to other standard loss, i.e. Dice and CE, the proposed loss does not provide good results and improve with a tiny gap.

**Summary Of The Paper:**

The paper provides an explicit bound relationship an information theoretic analysis, which uncover hidden label-marginal biases
The paper proposes LABEL-MARGINAL BIASES which combines CE loss with KL divergence or L1 norm to address class imbalance but without losing generality.

The evaluation is conducted on Retina dataset and Cityscapes.

**Summary Of The Review:**

Novelty
Clarity
Significance

---

> ### Author Response · Authors · 2021-11-23
> **Response to Reviewer 3tvn**
>
> We thank the reviewer for the time spent on our paper. We should mention that we are completely disappointed about the short/non-informative/misleading review of our work, with the three-word review summary; the many incorrect claims about the work (we provide details below); the unfounded criticism about the lack of novelty (no prior-art references given); and the dismissal of a widely used public retinal data set (in which the ground-truth segmentations were built by 45 experts and are oval only in a few samples).
>
> **1. The paper claim to address imbalanced data. It is not clear how to regularization term address that problem. Many recent loss functions for imbalanced data are not included in the paper.**
>
> We emphasize that the objective of the paper is not to propose a yet another loss for extremely imbalanced segmentation. We tackle the more general problem where class balance varies significantly across training samples, and challenge the common belief in the literature that widely used Dice and CE are complementary/different. Our main contribution is the technical insights that connect tightly CE and Dice and show that their main difference is essentially in their label-marginal biases, which explains why composite CE-Dice losses yield state-of-the-art performances [1], and why our label-marginal regularization is highly competitive. Furthermore, we refer Reviewer 3tvn to the very recent review and comprehensive experimental study in [1], where authors perform evaluations on more than 20 segmentation losses in a variety of class imbalance scenarios. In particular, [1] found that composite CE-Dice losses consistently obtained the best performances across all the settings. Also, as stated in our paper, [1] discusses how most segmentation losses are variants of CE and Dice, or a combination of both. This motivated our experimental comparison with highly competitive CE-Dice composite losses. It is noteworthy to mention that losses specifically designed to tackle the extreme imbalance case, e.g., focal or Tversky loss, do not outperform the aforementioned CE-Dice losses [1]. Furthermore, other popular losses such as Boundary loss (also included in the evaluations in [1]) works only when used jointly with the Dice loss. Our theoretical analysis challenges the wide use of Dice within the medical-imaging community, and points to a simple but non-trivial correction of CE as an effective alternative.
>
> **2. The ground-truth on retinal lesions dataset is not reliable because the lesion is shown in an oval.**
>
> This statement is incorrect, and dismisses the hard work of a panel of 45 experts in retinal images [2], who annotated about 35k lesions in over 1.5k retinal images from the widely used Kaggle retinal image dataset, with each image being assigned to at least three annotators [2]. We have included additional images in the appendix of the revised version, where the ground-truth segmentations are not elliptical (Sec. G). We believe that mentioning that the ground truth is not reliable is a strong and unfounded statement by the reviewer (unless the reviewer has more experience than the panel of retinal imaging experts responsible of the annotations).
>
> **3. The proposed loss provides similar performance with other well known losses.**
>
> We kindly disagree with the reviewer. The proposed loss, regardless of the form of the regularization term, brings significant improvements over the existing losses as evidenced by the result Tables (up to 4 percentage points in HD). This includes comparison with state-of-the-art composite loss CE-Dice [1]. Furthermore, these improvements are consistent across the backbones employed in our experiments.
>
> **4. The best performance on each data will depend on the choice of $\lambda$ value.**
>
> This is the case of all state-of-the-art composite losses [1], including CE-Dice and boundary loss mentioned by the reviewer (which has to be used with Dice). The optimal weighting factor $\lambda$ is chosen over a validation set for all competing composite losses, in a fair way.
>
> **5. There is no ablation study on imbalanced data or training stability.**
>
> We are confused by the "ablation study on imbalanced data". Does it refer to adding more datasets in the experiments?
> If so, we believe our extensive experiments on one medical image dataset and one natural dataset is convincing to confirm the theoretical analysis in this paper, as well as the effect of our method under different scenarios and networks. Also, it is unclear what is exactly meant by "ablation on training stability"
>
> **References**
>
> [1] Ma et al., MedIA 2021. Loss odyssey in medical image segmentation.
>
> [2] Wei et al., ICPR 2020. Learn to segment retinal lesions and beyond.

---

### Author Response · Authors · 2021-11-23
**To AC and all reviewers (summary of revisions)**

We thank all 4 reviewers for their efforts, and are pleased that 2 reviewers, R2 (P7HM) and R3 (QsrX), found the theoretical analysis sound and the insights important to the community. The criticisms of R4 (njoP) are mainly about ablation studies on the balancing weight, additional experimental results and questions about the quality of the dataset used, which we address in the revision and clarify in the detailed responses below. We kindly disagree with R4 (njoP) that the main message conveyed by our Propositions 1 and 2 (i.e., Dice and CE are closely related and essentially differ by their label-marginal bias) is intuitive and common knowledge within the community. As evidence that this is not the case: It is widely argued within the medical imaging community that Dice and CE are complementary losses, which motivated the use of compound CE-Dice losses; see, for instance, the recent comprehensive review and experiments in [1]. This is also confirmed by the fact that Reviewer QsrX requested further clarifications of the non-trivial technical result in Proposition 1.

As for the review quality from R1 (3tvn), we should mention that we are completely disappointed about the short/non-informative/misleading review of our work, the many INCORRECT claims about the work (we provide details below in our answer to Reviewer 3tvn), the unfounded criticism about the lack of novelty (no prior-work references  or arguments given) and the dismissal of a widely used public retinal data set (in which a panel of 45 experts annotated about 35k lesions on over 1.5k retinal images from a widely used Kaggle retinal image dataset [2], and where the ground-truth segmentations are oval only in a few samples). Furthermore, we kindly point out here that it is unusual that reviewer 3tvn is releasing the review publicly and not in the appropriate official review section. Therefore, we kindly ask the Area Chair and other reviewers to dismiss the misleading/non-informative comments by Reviewer 3tvn.

We hope the revised version, along with the detailed responses below, address the main concerns and clarify further the contribution.
Here following a summary of the changes we made in the revision:

* We added results with more backbones (i.e., Res34-FPN, Res50-FPN) on Retinal Lesion Dataset, as shown in **Table 1**. These results consistently validate the theoretic analysis and the effect of the proposed regularizers.

* In the ablation study regarding the balancing weight (please refer to the revised paragraph *On the balancing weight in the composite losses* in the *Experiments* section), we include a discussion about CE-log(Dice). We clarified that we also optimize the balancing weight for it and report the best value over validation, for fair comparisons with state-of-the-art composite losses. A detailed ablation study on the balancing weight of CE-log(Dice) is also added in **Fig. E.1** of the appendix.

* In Table 4, we add the scores of -log(Dice) on Cityscapes test set, showing a consistent performance.

* **We clarified the annotation quality of the Retinal Lesions dataset.** We mentioned in the *Datasets* paragraph, under the *Experiments* section, that a panel of 45 experienced ophthalmologist was formed to label this dataset and each image was assigned to at least three annotators to get a trust-worthy pixel-level annotation. In addition, we showcase more examples in **Fig. G.1** of the appendix, where segmentations are not oval.

**References**

[1] Ma et al., MedIA 2021. Loss Odyssey in Medical Image segmentation.

[2] Wei et al., ICPR 2020. Learn to segment retinal lesions and beyond.

---

### Decision · Program_Chairs · 2022-01-20

**Decision:**

Reject

**Comment:**

The submission evaluates the relationship between (logarithmic) Dice loss and cross-entropy loss, arguing for a similar decomposition into ground truth and "hidden label-marginal biases."  The submission received mixed reviews, with two reviewers voting for rejection, and two feeling that it is marginally above the acceptance threshold.  Setting aside the numerical scores, there are reasons to believe that this submission, while interesting, has shortcomings that limit its relevance to the wider ICLR community.  These include
- Very many losses have been proposed for imbalanced classification / (medical) image segmentation, such as Jaccard and Tversky index or ranking measures, although admittedly Dice is probably the most popular in the medical imaging literature due to historic reasons.  Arguably, Dice is less well-behaved from a theoretical perspective compared to other options (e.g. it does not even form a metric), and may not be the most relevant point of departure for a representation learning conference.  The literature review misses many relevant papers on such losses, including papers that specifically are focused on the relationship between Dice and cross-entropy, e.g. Eelbode et al., IEEE-TMI 2020 and citations therein.
- The empirical results do not show substantially improved results compared to baselines.

On the balance, this does not cross the threshold for acceptance to a competitive venue such as ICLR.